# A bacterial effector protein prevents MAPK-mediated phosphorylation of SGT1 to suppress plant immunity

Gang Yu[1], Liu Xian[1,2], Hao Xue[1,2], Wenjia Yu[1,2], Jose S. Rufian[1], Yuying Sang[1], Rafael J. L. Morcillo[1], Yaru Wang[1,2], Alberto P. Macho[1]*

1 Shanghai Center for Plant Stress Biology, CAS Center for Excellence in Molecular Plant Sciences, Chinese Academy of Sciences, Shanghai, China, 2 University of Chinese Academy of Sciences, Beijing, China

* alberto.macho@psc.ac.cn, alberto.macho@icloud.com

**Data Availability Statement:** All relevant data are within the manuscript and its Supporting Information files.

## Abstract

Nucleotide-binding domain and leucine-rich repeat-containing (NLR) proteins function as sensors that perceive pathogen molecules and activate immunity. In plants, the accumulation and activation of NLRs is regulated by SUPPRESSOR OF G2 ALLELE OF *skp1* (SGT1). In this work, we found that an effector protein named RipAC, secreted by the plant pathogen *Ralstonia solanacearum*, associates with SGT1 to suppress NLR-mediated SGT1-dependent immune responses, including those triggered by another *R. solanacearum* effector, RipE1. RipAC does not affect the accumulation of SGT1 or NLRs, or their interaction. However, RipAC inhibits the interaction between SGT1 and MAP kinases, and the phosphorylation of a MAPK target motif in the C-terminal domain of SGT1. Such phosphorylation is enhanced upon activation of immune signaling and contributes to the activation of immune responses mediated by the NLR RPS2. Additionally, SGT1 phosphorylation contributes to resistance against *R. solanacearum*. Our results shed light onto the mechanism of activation of NLR-mediated immunity, and suggest a positive feedback loop between MAPK activation and SGT1-dependent NLR activation.

## Author summary

Plant immune receptors are subjected to a tight regulation in order to avoid auto-immune responses, but must be promptly activated upon perception of pathogen threats. Plant intracellular immune receptors can perceive effector proteins injected inside plant cells; for most of these receptors, the SUPPRESSOR OF G2 ALLELE OF *skp1* (SGT1) constitutes an essential regulator that controls their stability and activation. In this work, we found that phosphorylation mediated by MAPKs regulates SGT1, and is required for the activation of intracellular immune receptors. Supporting the biological relevance of this finding, we found that a bacterial pathogen, *Ralstonia solanacearum*, injects an effector protein that inhibits the interaction between MAPKs and SGT1, thereby suppressing SGT1 phosphorylation and the associated effector-triggered immune responses. These results suggest the existence of a phosphorylation-mediated feedback loop between

**Funding:** This work was supported by the Strategic Priority Research Program of the Chinese Academy of Sciences (grant XDB27040204), the National Natural Science Foundation of China (NSFC; grant 31571973), the Chinese 1000 Talents Program, and the Shanghai Center for Plant Stress Biology (Chinese Academy of Sciences) to APM. GY was partially supported by the China Postdoctoral Science Foundation (grant 2016M600339). The funders had no role in study design, data collection and analysis, decision to publish, or preparation of the manuscript.

**Competing interests:** The authors have declared that no competing interests exist.

MAPKs and SGT1, which leads to the activation of immune responses, and can be targeted by bacterial virulence activities.

## Introduction

The activation and suppression of plant immunity are key events that determine the outcome of the interaction between plants and bacterial pathogens. When in close proximity to plant cells, conserved bacterial molecules (termed PAMPs, for pathogen-associated molecular patterns) may be perceived by receptors at the surface of plant cells, eliciting the activation of the plant immune system. This leads to the establishment of PAMP-triggered immunity (PTI) [1], which restricts bacterial growth and prevents the development of disease. Most bacterial pathogens have acquired the ability to deliver proteins inside plant cells via a type-III secretion system (T3SS); such proteins are thus called type-III effectors (T3Es) [2]. Numerous T3Es from adapted bacterial plant pathogens have been reported to suppress PTI [3]. This, in addition to other T3E activities [4], enables bacterial pathogens to proliferate and cause disease. Resistant plants have evolved mechanisms to detect such bacterial manipulation through the development of intracellular receptors defined by the presence of nucleotide-binding sites (NBS) and leucine-rich repeat domains (LRRs), thus termed NLRs [5]. The detection of T3E activities through NLRs leads to the activation of immune responses, which effectively prevent pathogen proliferation [6]. The outcome of these responses is named effector-triggered immunity (ETI), and, in certain cases, may cause a hypersensitive response (HR) that involves the collapse of plant cells. In an evolutionary response to this phenomenon, T3E activities have evolved to suppress ETI [7], which in turn exposes bacterial pathogens to further events of effector recognition. For these reasons, the interaction between plants and microbial pathogens is often considered an evolutionary 'arms race', where the specific combination of virulence activities and immune receptors in a certain pathogen-plant pair ultimately defines the outcome of the interaction [7]. Although the suppression of PTI by T3Es has been widely documented [3], reports about T3Es that suppress ETI, and their biochemical characterization, remain scarce.

Interestingly, PTI and ETI signaling often involve shared regulators [8], such as mitogen-activated protein kinases (MAPK). The activation of MAPKs is one of the earliest signaling events upon perception of both PAMPs and effectors [9], and is essential for the activation of immune responses. Although sustained MAPK activation has been reported to be critical for the robust development of ETI [10], the signaling events and phosphorylation substrates that link early MAPK activation upon pathogen recognition and downstream MAPK activation as a consequence of NLR activation remain unknown.

The study of microbial effectors has become a very prolific approach to discover and characterize components of the plant immune system and other cellular functions that play important roles in plant-microbe interactions [11]. In this regard, T3E repertoires constitute a powerful tool for researchers to understand plant-bacteria interactions. The *Ralstonia solanacearum* species complex (RSSC) groups numerous bacterial strains able to cause disease in more than 250 plant species, including important crops, such as tomato, potato, and pepper, among others [12, 13]. *R. solanacearum* invades the roots of host plants, reaching the vascular system and colonizing xylem vessels in the whole plant, eventually causing their blockage and subsequent plant wilting [14, 15]. It has been speculated that the versatility and wide host range of *R. solanacearum* is associated to its relatively large T3E repertoire in comparison with other bacterial plant pathogens; the reference GMI1000 strain serves as an example, being able to secrete more than 70 T3Es [16]. Among these T3Es, RipAC (also called PopC) [16, 17] is

conserved in most *R. solanacearum* sequenced to date, and is therefore considered a core T3E in the RSSC (S1A and S1B Fig) [16]. Domain prediction for the RipAC amino acid sequence reveals the presence of an LRR domain and no other predicted enzymatic domain (S1C Fig). Interestingly, it has been shown that RipAC from different *R. solanacearum* strains is promptly expressed upon bacterial inoculation in different host plants [18, 19] suggesting a role in the establishment of the interaction at early stages of the infection process, but the specific virulence activity and plant target(s) of RipAC remain unknown. In this work, we found that RipAC contributes significantly to *R. solanacearum* virulence. RipAC associates with the major ETI regulator SGT1 to suppress NLR-mediated immune responses, including those triggered by another *R. solanacearum* effector, RipE1. We found that RipAC inhibits the interaction between SGT1 and MAP kinases, and the phosphorylation of a MAPK target motif in the C-terminal domain of SGT1 associated to the activation of ETI. We also showed that SGT1 phosphorylation contributes to resistance against *R. solanacearum*, and this is particularly evident in the absence of RipAC. Altogether, our results highlight a novel mechanism for effector suppression of plant immunity, and reveal a potential feedback loop between the activation of MAPKs and SGT1-mediated regulation of ETI.

## Results

### RipAC contributes to *Ralstonia solanacearum* infection

It is well established that T3Es collectively play an essential role in the development of disease caused by most bacterial plant pathogens, including *R. solanacearum* [20]. To study the contribution of RipAC to *R. solanacearum* infection, we generated a *ΔripAC* mutant strain using the reference phylotype I GMI1000 strain as background (S1D Fig), and confirmed that the growth of this strain in rich laboratory medium is indistinguishable from the wild-type (WT) strain (S1D–S1G Fig). We then performed soil-drenching inoculation of Arabidopsis Col-0 WT plants with both WT and *ΔripAC* strains, and monitored the development of wilting symptoms associated to disease. Compared with plants inoculated with WT GMI1000, plants inoculated with the *ΔripAC* mutant displayed attenuated disease symptoms (Fig 1A and 1B, S2A Fig). This attenuation was complemented by the expression of *ripAC*, driven by its native promoter, in the *ΔripAC* mutant background (Fig 1A and 1B, S2A and S1E–S1G Figs), confirming that virulence attenuation was indeed caused by the lack of the *ripAC* gene. Given the intrinsic complexity and biological variability of the soil-drenching inoculation assay, we did not detect a significant attenuation every time we performed this experiment (S2A Fig), suggesting that the attenuation caused by the absence of RipAC is near the limit of detection of this assay. Tomato is a natural host for several *R. solanacearum* strains [21]. As observed in Arabidopsis, tomato plants inoculated with the *ΔripAC* strain showed delayed and/or reduced symptom development in most experiments (Fig 1C and 1D, S2B Fig). This attenuation was complemented by the expression of *ripAC* in the mutant background (Fig 1C and 1D, S2B Fig).

In order to increase the accuracy of our virulence assays and to determine whether the reduced disease symptoms correlate with reduced bacterial multiplication inside host plants, we set up an assay to quantify bacterial numbers in the xylem sap of tomato plants upon injection of the bacterial inoculum in the stem. Compared to plants inoculated with WT GMI1000 or the complementation strain, plants inoculated with the *ΔripAC* strain showed reduced bacterial numbers (Fig 1E). Altogether, these results suggest that, despite the large effector repertoire of *R. solanacearum* GMI1000, RipAC plays a significant role during the infection process in different host plants.

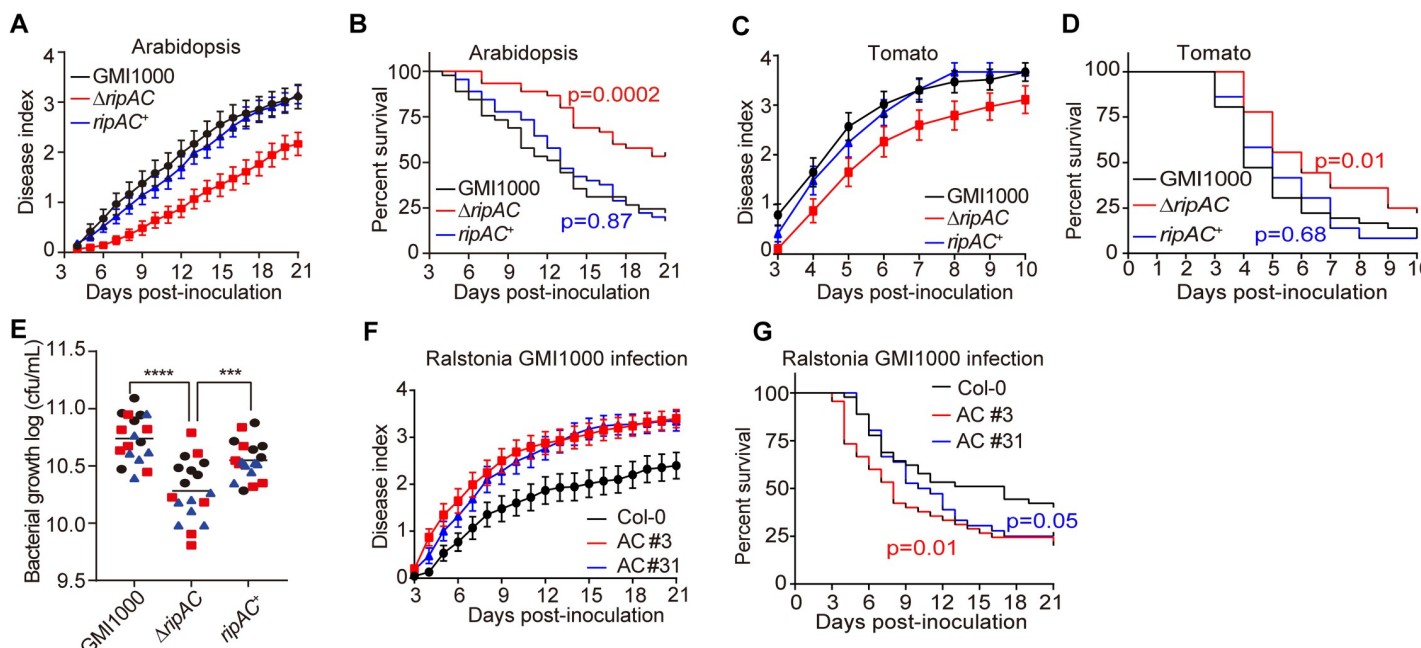

**Fig 1. RipAC contributes to *Ralstonia solanacearum* virulence in Arabidopsis and tomato.** (A, B) Soil-drenching inoculation assays in Arabidopsis were performed with GMI1000 WT, *ΔripAC* mutant, and RipAC complementation (*ripAC*+) strains. In (A) the results are represented as disease progression, showing the average wilting symptoms in a scale from 0 to 4. Values from 3 independent biological repeats were pooled together (mean ± SEM; n = 45). Curves for each replicate are shown in S2A Fig. (B) Survival analysis of the data in (A); the disease scoring was transformed into binary data with the following criteria: a disease index lower than 2 was defined as '0', while a disease index equal or higher than 2 was defined as '1' for each specific time point. Statistical analysis was performed using a Log-rank (Mantel-Cox) test (n = 45 for each strain), and the corresponding p value is shown in the graph with the same colour as each curve. (C, D) Soil-drenching inoculation assays in tomato were performed with GMI1000 WT, *ΔripAC* mutant, and RipAC complementation (*ripAC*+) strains. In (C) and (D) the analyses were performed the same as in (A) and (B) (n = 36 for each strain), and the corresponding p value is shown in the graph in (D) with the same colour as each curve. Curves for each replicate are shown in S2B Fig. (E) Growth of *R. solanacearum* GMI1000 WT, *ΔripAC* mutant, and RipAC complementation (*ripAC*+) strains in tomato plants. 5 μL of bacterial inoculum ($10^6$ cfu mL$^{-1}$) were injected into the stem of 4-week-old tomato plants and xylem sap was taken from each infected plants for bacterial titer calculation 3 days post-inoculation (dpi). Different colours represent values obtained in 3 independent biological repeats, and horizontal bars represent average values (n = 6 plants per strain in each repeat). Asterisks indicate significant differences (*** $p<0.001$, **** $p<0.0001$, *t*-test). (F, G) Soil-drenching inoculation assays in Col-0 WT and RipAC-expressing Arabidopsis lines (AC #3, AC #31, two independent lines) with GMI1000 WT strain. The analyses were performed the same as in (A) and (B) (n = 45 for each genotype), and the corresponding p value is shown in the graph in (G) with the same colour as each curve. Curves for each replicate are shown in S3C Fig.

To complement our genetic analysis of the importance of RipAC in promoting susceptibility to *R. solanacearum*, we generated stable Arabidopsis transgenic lines expressing RipAC from a single 35S constitutive promoter (S3A and S3B Fig). Independent homozygous transgenic lines expressing RipAC showed enhanced disease symptoms upon soil-drenching inoculation with *R. solanacearum* (Fig 1F and 1G, S3C Fig), indicating that RipAC promotes susceptibility to *R. solanacearum* from within plant cells.

## RipAC interacts with SGT1 from different plant species

Upon transient expression in *Nicotiana benthamiana*, RipAC fused to a C-terminal GFP tag shows a sharp signal at the cell periphery of leaf epidermal cells (S4A Fig). Fractionation of protein extracts from *N. benthamiana* expressing RipAC, showed abundant RipAC in both microsomal and cytosolic fractions (S4B Fig). To identify plant proteins that physically associate with RipAC in plant cells, we performed immunoprecipitation (IP) of RipAC-GFP using GFP-trap agarose beads, and analyzed its interacting proteins using liquid chromatography coupled to tandem mass-spectrometry (LC-MS/MS). Together with RipAC-GFP immunoprecipitates, we identified peptides of NbSGT1 (S5A Fig), homolog of the Arabidopsis SUPPRESSOR OF G2 ALLELE OF *skp1* (SGT1), a protein required for the induction of disease

resistance mediated by most NLRs [22, 23]. The localization of both isoforms of SGT1 in Arabidopsis (AtSGT1a and AtSGT1b) has been reported to be nucleocytoplasmic [24]. When we expressed SGT1 (NbSGT1, AtSGT1a, or AtSGT1b) fused to a C-terminal RFP tag, we also detected an apparently cytoplasmic localization, with occasional nuclear localization, although we also detected co-localization with the plasma membrane marker CBL-GFP [25] (S5B and S5C Fig). Interestingly, we detected co-localization of RipAC-GFP and NbSGT1-RFP, AtSGT1a-RFP, or AtSGT1b-RFP (S5D Fig). These results also suggest that RipAC does not affect SGT1 subcellular localization. To confirm the physical interaction between RipAC and SGT1 from different plant species, we co-expressed RipAC-GFP with either NbSGT1, AtSGT1a, AtSGT1b or the tomato ortholog of SGT1b (SlSGT1b). Targeted IP of RipAC-GFP showed CoIP of all different SGT1 orthologs upon transient co-expression in *N. benthamiana* (Fig 2A). Similarly, split-luciferase complementation (Split-LUC) assays in *N. benthamiana* showed luciferase signal upon expression of RipAC with AtSGT1a, AtSGT1b, NbSGT1 or SlSGT1b fused to different luciferase halves (Fig 2B and S5E Fig). We also performed split-YFP assays, showing reconstitution of the YFP signal at the cell periphery upon expression of RipAC and AtSGT1a, AtSGT1b or NbSGT1 tagged with one of the portions of a split-YFP molecule (S5F Fig), indicating a direct interaction of these proteins at the cell periphery. The direct interaction between RipAC and AtSGT1a/b was also confirmed by *in vitro* pull-down (Fig 2C). Altogether, these results indicate that RipAC physically associates with SGT1 from different species inside plant cells.

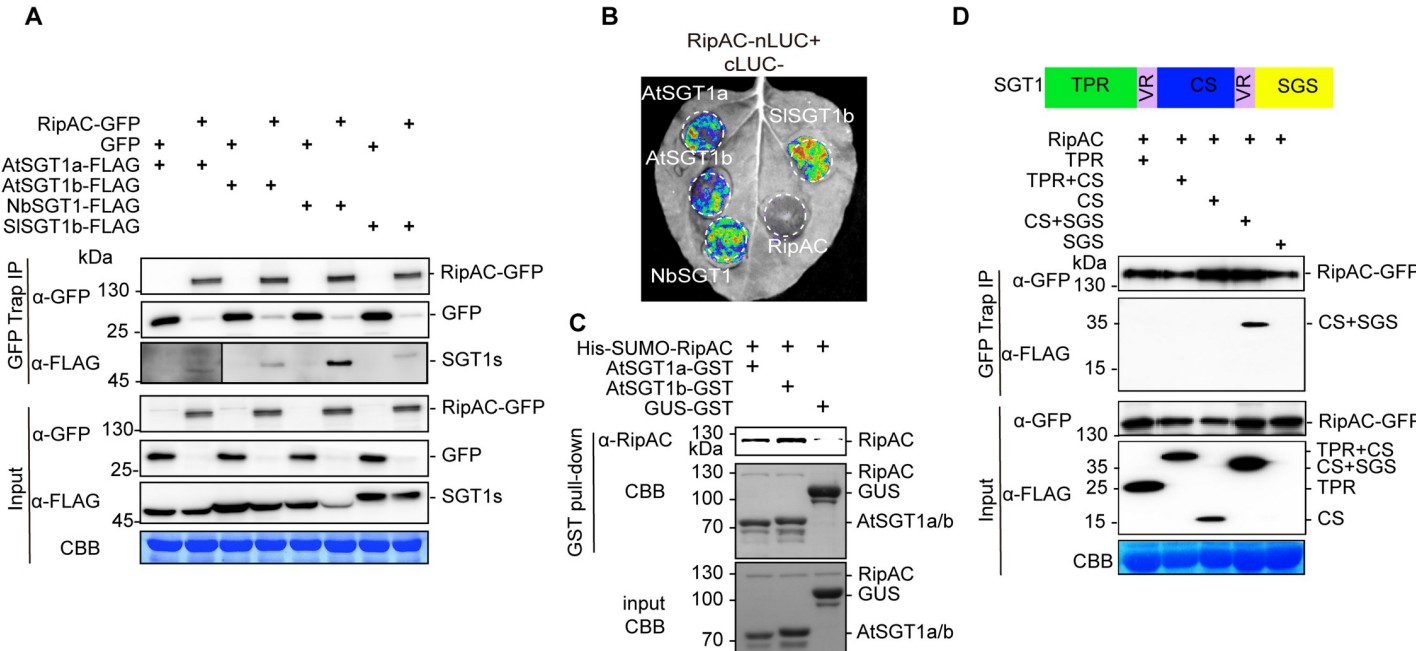

**Fig 2. RipAC associates with SGT1 in plant cells.** (A) CoIP to determine interactions between RipAC and SGT1s transiently expressed in *Nicotiana benthamiana*. The signal in the interaction between RipAC and AtSGT1a was weak and is shown with longer exposure. (B) Split-LUC assay to determine direct interaction between RipAC and SGT1 transiently expressed in *N. benthamiana*. (C) Pull-down assay to determine direct interaction between RipAC and SGT1a/b. His-tagged RipAC was incubated with immobilized GST-AtSGT1a/AtSGT1b/GUS. After 4 rounds of wash, bound proteins were eluted and subjected to western blot analysis using anti-RipAC antibody. Coomassie blue (CBB) staining showed the visualization of both input and GST pull-down proteins. (D) CoIP to determine interactions between RipAC and different truncated versions of NbSGT1 in *N. benthamiana*. The diagram summarizes the different domains of SGT1: TPR, N-terminal tetratricopeptide repeat (TPR) domain; CS, CHORD-SGT1 domain; SGS, C-terminal SGT1-specific domain; VR, variable region. The experiments in (A-C) were repeated at least 3 times with similar results. The experiment in (D) was repeated 2 times with similar results. In western blot assays, protein marker sizes are provided for reference. In (A) and (D), blots were stained with Coomassie Brilliant Blue (CBB) to verify equal loading.

SGT1 is conserved in eukaryotes, and has three defined domains: an N-terminal tetratrico-peptide repeat (TPR) domain, a CHORD-SGT1 (CS) domain, and a C-terminal SGT1-specific (SGS) domain [26]. In order to determine which NbSGT1 domain is required for RipAC inter-action, we expressed different combinations of these domains (Fig 2D) bound to a FLAG tag, and tested their interaction with immunoprecipitated RipAC-GFP. Although we were unable to express the SGS domain alone, we detected interaction between RipAC and a truncated SGT1 protein containing the CS+SGS domain (Fig 2D). The CS domain alone or other SGT1 domains did not interact with RipAC, ruling out interaction artifacts due to protein overex-pression, and suggesting that RipAC associates specifically with the SGS domain, although it cannot be ruled out that RipAC requires both CS+SGS domains to associate with SGT1. Inter-estingly, the SGS domain is required for all the known SGT1b functions in immunity [24, 26].

SGT1 physically associates with several NLRs [27–29]; here we found that AtSGT1b also interacts with the NLR RPS2 [30] (S5G Fig). To determine whether RipAC interferes with the interaction between SGT1 and NLRs, we performed Split-LUC assays between AtSGT1b and RPS2 in the presence of RipAC. Interestingly, RipAC did not seem to affect RPS2 accumula-tion or the interaction between AtSGT1b and RPS2 (S5G and S5H Fig). Although this result does not completely rule out that RipAC may quantitatively interfere with SGT1-NLR interac-tions, it suggests that this is not the main activity underlying RipAC targeting of SGT1.

## RipAC inhibits SGT1-mediated immune responses in *N. benthamiana* and Arabidopsis

To determine whether RipAC has the potential to suppress SGT1-mediated immune responses, we first performed transient expression of RipAC-GFP in *N. benthamiana*, and monitored the induction of cell death associated to the hypersensitive response (HR) devel-oped as part of an ETI response, such as that triggered by overexpression of the NLR-encoding gene *RPS2* [31] and the expression of the oomycete effector Avr3a together with the cognate NLR R3a [32]. RipAC suppressed the HR triggered by RPS2 and Avr3a/R3a, showing its ability to suppress ETI. We have recently reported that the *R. solanacearum* T3E RipE1 triggers SGT1-dependent immunity in *N. benthamiana* [33], and, accordingly, RipAC was also able to suppress RipE1-induced HR (Fig 3A). Interestingly, RipAC was not able to suppress the cell death triggered by other elicitors, such as Bax [34] and the oomycete elicitor INF1 [35] (Fig 3B), suggesting that RipAC is not a general suppressor of cell death. Ion leakage measurements validated that RipAC suppresses SGT1-dependent ETI-associated cell death (e.g. triggered by RPS2 or RipE1), but had a minor impact on BAX-triggered cell death (S6A and S6B Fig). It is noteworthy that the RipAC inhibition of HR was not caused by an inhibition of Agrobacter-ium-mediated transient expression (Fig 3C and 3E).

The activation of NLRs is known to trigger the activation of mitogen-activated protein kinases (MAPKs) [10, 36]. Consistently, transient overexpression of RPS2 in *N. benthamiana* leads to an activation of MAPKs (SIPK and WIPK) that precedes the onset of ion leakage and macroscopic cell death (S6A–S6D Fig). In our transient expression assays, RipAC significantly suppressed the activation of MAPKs triggered by overexpression of RPS2, Avr3a/R3a, or by the expression of RipE1 (Fig 3C–3F), supporting the notion that RipAC suppresses early steps in ETI activation rather than downstream responses that lead to cell death.

To determine whether RipAC inhibits the final outcome of ETI (i.e. restriction of bacterial proliferation), we inoculated RipAC-expressing Arabidopsis transgenic lines (S3A and S3B Fig) with *Pto* DC3000 expressing different T3Es that activate NLR-dependent immunity, namely AvrRpm1 [37], AvrRpt2 [30], or AvrRps4 [38]. Such strains, compared to *Pto* DC3000 carrying an empty vector (EV) (Fig 3G), show deficient replication in Arabidopsis, due to

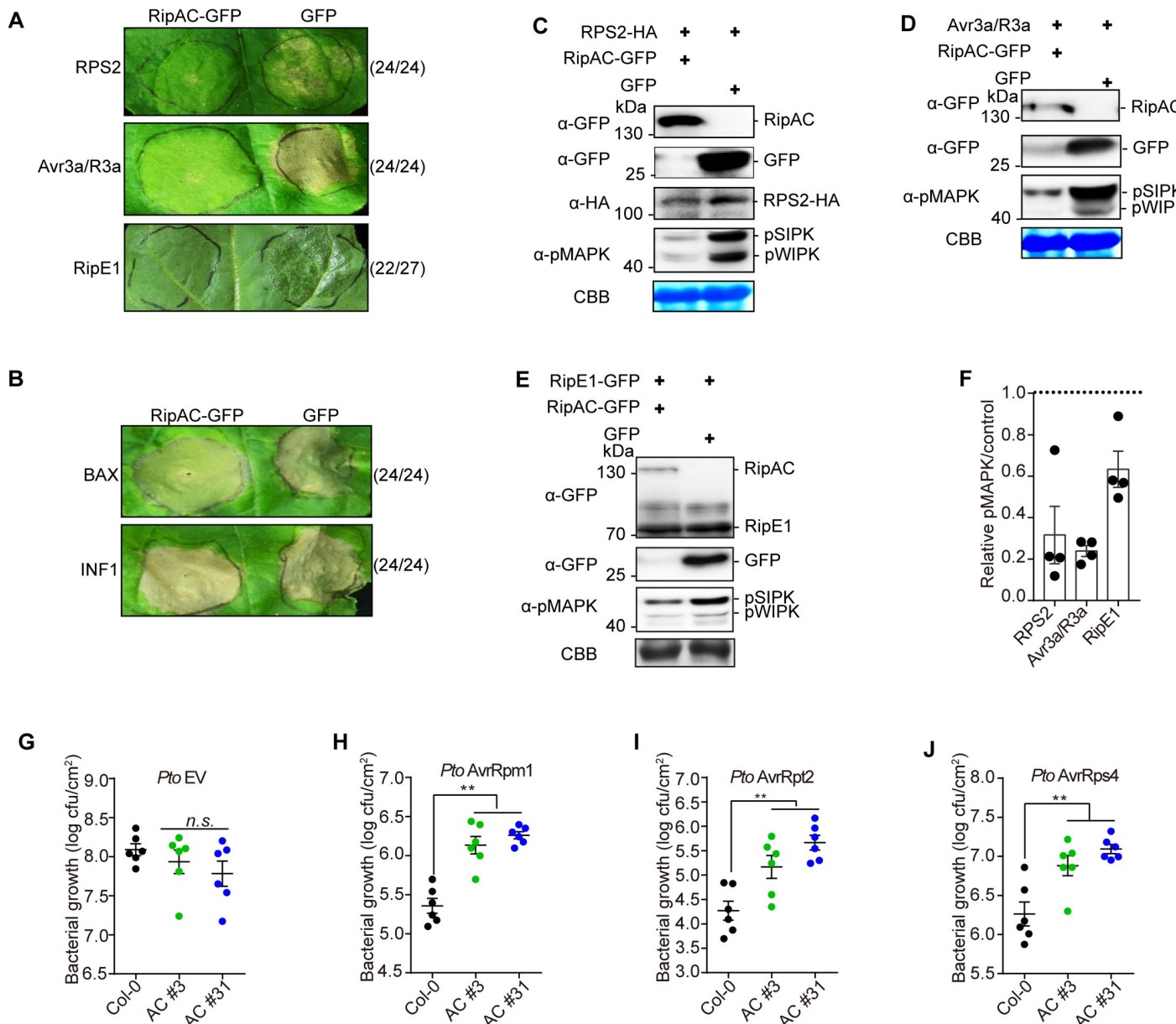

**Fig 3. RipAC inhibits SGT1-dependent immune responses.** (A) RipAC suppresses RPS2-, Avr3a/R3a-, or RipE1-associated cell death in *Nicotiana benthamiana*. (B) RipAC does not suppress BAX- or INF1-induced cell death in *N. benthamiana*. In (A) and (B) Photographs were taken 5 days after the expression of the cell death inducer. The numbers beside the photographs indicate the ratios of infiltration showing the presented result among the total number of infiltrations. (C-F) RipAC suppresses MAPK activation triggered by the overexpression of RPS2 (C), Avr3a/R3a (D), or RipE1 (E) in *N. benthamiana*. Plant tissues were taken one day after cell death inducer expression. (F) Quantification of relative pMAPK in (C-E). The quantification of the pMAPK band in RipAC expression spots is represented relative to the intensity in the GFP control in each assay (mean ± SEM of 4 independent biological replicates). (G-J) RipAC suppresses effector-triggered immunity in Arabidopsis. Leaves of Col-0 WT and RipAC-transgenic lines (AC #3 and AC #31) were hand-infiltrated with *Pto* empty vector (EV, G), *Pto* AvrRpm1 (H), *Pto* AvrRpt2 (I), or *Pto* AvrRps4 (J). Samples were taken 3dpi (mean ± SEM, n = 6, ** $p < 0.01$, *t*-test). These experiments were repeated at least 3 times with similar results. In western blot assays, protein marker sizes are provided for reference. Blots were stained with Coomassie Brilliant Blue (CBB) to verify equal loading.

immune responses triggered by these T3Es (Fig 3H–3J). However, transgenic lines expressing RipAC showed enhanced susceptibility to these strains (Fig 3H–3J), indicating that RipAC is able to effectively suppress the final outcome of NLR-mediated immunity (i.e. the restriction of bacterial growth) in Arabidopsis.

## SGT1 is phosphorylated in a MAPK target motif upon activation of ETI

Since post-translational modifications (PTMs) have been shown to impact SGT1 function [39–41], we sought to determine whether RipAC causes any alteration of SGT1 PTMs in plant cells. To this end, we co-expressed NbSGT1 fused to a FLAG tag together with RipAC (or a GFP control) in *N. benthamiana* leaves, and then performed IP using an anti-FLAG resin followed by LC-MS/MS. Among the IPed SGT1 peptides, we detected two peptides, located in the SGS domain, containing phosphorylated serine residues (S282 and S358) (S7 Fig). Interestingly, a phosphorylable residue is conserved in these positions among plants from different species (S8A Fig), and S358 is embedded in a canonical MAPK-mediated phosphorylation motif conserved across the plant kingdom (S8B Fig). Interestingly, AtSGT1b has been shown to physically interact with MAPK4 [42], and SGT1 from tobacco (NtSGT1) S358, equivalent to T346 in AtSGT1b, is directly phosphorylated by NtSIPK [39]. By performing targeted CoIP, we found that both AtSGT1a and AtSGT1b associate with AtMAPK4 and AtMAPK6, but not with AtMAPK3, which showed very low accumulation in this assay (Fig 4A). In Split-LUC assays, we found direct interaction of both AtSGT1a and AtSGT1b with AtMAPK3 and AtMAPK4, but not with AtMAPK6 (Fig 4B and 4C, S9 Fig). These results indicate that AtMAPK4 interacts with AtSGT1a and AtSGT1b. While they also suggest an interaction of AtSGT1s with AtMAPK3 and AtMAPK6, different results were obtained in different interaction assays, probably due to the different experimental conditions. To determine whether MAPK-mediated SGT1 phosphorylation is enhanced upon activation of ETI, we raised an antibody against the potential MAPK phosphorylated peptide, conserved in both AtSGT1a and AtSGT1b, containing phosphorylated T346 (ESpTPPDGME), which specifically detected the phosphorylated peptide *in vitro* and *in planta* (S10A and S10B Fig). Using this antibody after *in vitro* kinase assays we observed that recombinant AtMAPK3 and AtMAPK6 directly phosphorylate recombinant AtSGT1a and AtSGT1b, and this phosphorylation was stronger in the presence of a constitutively active version of AtMKK5 (AtMKK5DD) (Fig 4D). Moreover, dexamethasone-induced expression of MKK5DD in Arabidopsis transgenic plants (Dex-MKK5DD) [43], which leads to constitutive activation of MAPK3/6, enhances SGT1 phosphorylation (Fig 4E and 4F). These results indicate that SGT1 undergoes MAPK-mediated phosphorylation.

Dexamethasone-induced expression of AvrRpt2 in Arabidopsis transgenic lines led to sustained MAPK activation and enhanced SGT1 phosphorylation (Fig 4G and 4H), suggesting that an enhancement of T346 phosphorylation occurs concomitantly to the activation of ETI. Interestingly, inoculation with *Pto* expressing AvrRpt2 leads to enhanced phosphorylation of SGT1 (Fig 4I and 4J) [10], and such enhancement was abolished in conditional mutants with abolished AtMAPK3 and AtMAPK6 activity in *Atmapk6* and *Atmapk3* mutant backgrounds, respectively [44, 45] (Fig 4I and 4J), indicating that SGT1 phosphorylation requires MAPK3 and MAPK6.

## MAPK-mediated phosphorylation contributes to SGT1 function in the activation of ETI

To determine the relevance of the phosphorylation of SGT1 S271/T346 for the activation of ETI, we generated single and double point mutants in these residues in AtSGT1b, substituting them for alanine residues (to abolish phosphorylation) or for aspartate residues (to mimic the negative charge associated to constitutive phosphorylation). Due to the difficulties in performing genetic analysis of SGT1, we used a transient approach in *N. benthamiana*, co-expressing the AtSGT1b variants together with RPS2. Alanine mutations (S271A/T346A: AtSGT1b AA) did not have a detectable impact on the RPS2-mediated cell death (Fig 5A), likely due to the

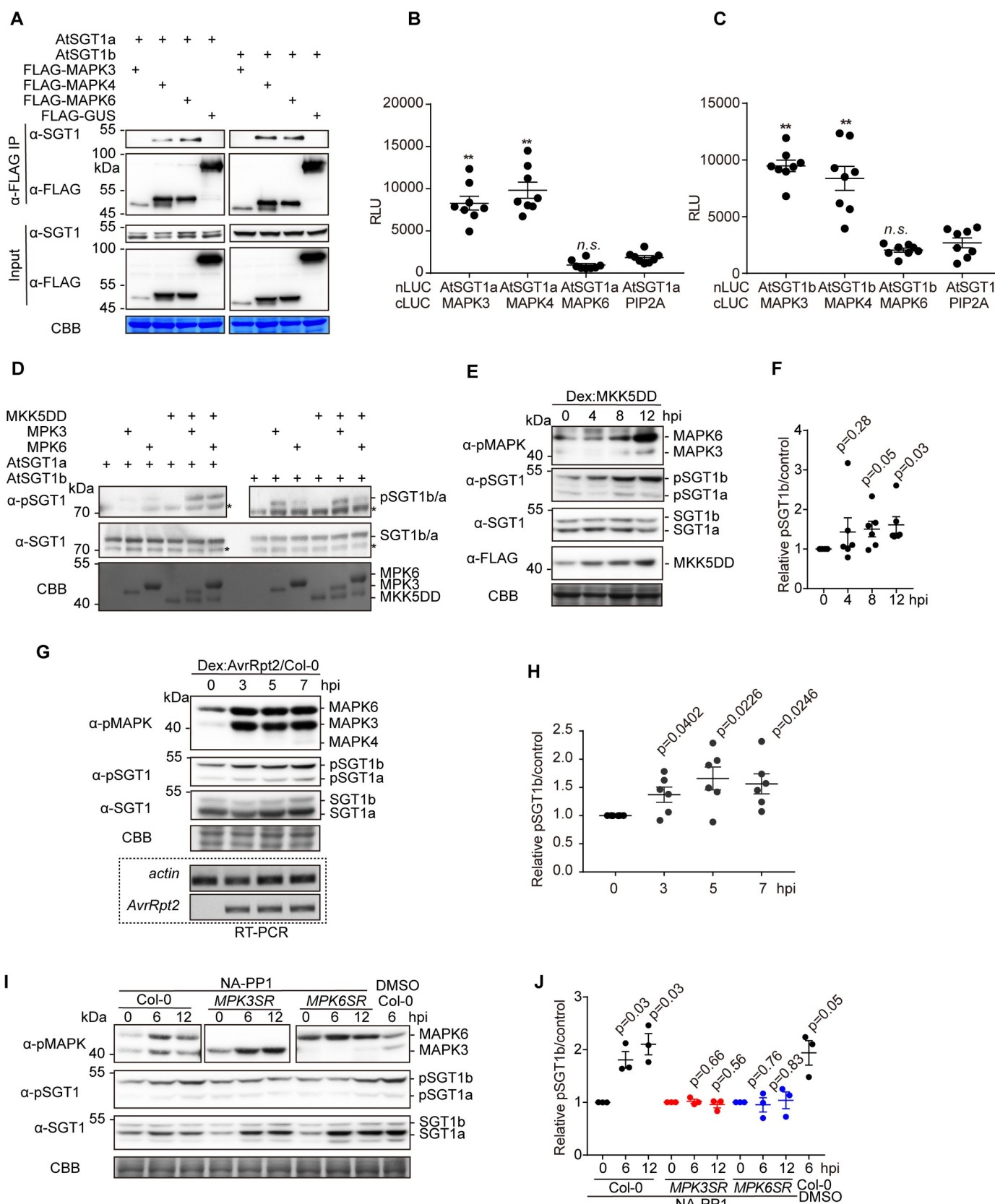

**Fig 4. MAPKs associate with SGTs in plant cells.** (A) AtSGT1a and AtSGT1b associate with MAPK4 and MAPK6 in *Nicotiana benthamiana*. CoIPs were performed as in Fig 2A. (B, C) AtSGT1a/b associate with AtMAPK3/4, but not with MAPK6, in Split-LUC assay in *N. benthamiana*. Agrobacterium combinations with different constructs were infiltrated in *N. benthamiana* leaves and luciferase activities were examined with microplate luminescence reader. The graphs show the mean value ± SEM (n = 8, ** p<0.01, *t*-test). The MAPK-PIP2A combination was used as negative control. (D) Phosphorylation of recombinant AtSGT1a/AtSGT1b by activated MPK3 and MPK6 in vitro. Recombinant MPK3/MPK6 were first activated by MKK5DD and subsequently incubated with AtSGT1a/AtSGT1b. Phosphorylated proteins (upper panel) were visualized by western blots using anti-SGT1 phosphorylation antibody, and the presence of SGT1 was visualized by western blots using anti-SGT1 antibody (middle panel). The equal input of MKK5DD and MPK3/MPK6 was confirmed by Coomassie blue (CBB) (lower panel). In all the blots, asterisks indicate non-specific bands. (E) MAPK activation triggered by MKK5DD expression correlates with increased AtSGT1 phosphorylation at T346 in Arabidopsis. Twelve-day-old Dex:MKK5DD plants were treated with 30 mM Dex, and samples were taken at the indicated time points. In (E), a custom antibody was used to detect the phosphorylated peptide containing AtSGT1 T346. The induction of MKK5DD was determined by western blot using anti-FLAG antibody. (F) Quantification of pSGT1b signal relative to Coomassie blue staining control in (E) (mean ± SEM of 6 independent biological repeats, p-values are shown, *t*-test). (G) MAPK activation triggered by AvrRpt2 expression correlates with increased AtSGT1 phosphorylation at T346 in Arabidopsis. Twelve-day-old Dex:AvrRpt2/Col-0 crossing F1 plants were treated with 30 mM Dex, and samples were taken at the indicated time points. In (G), a custom antibody was used to detect the phosphorylated peptide containing AtSGT1 T346. The induction of AvrRpt2 was determined by RT-PCR. (H) Quantification of pSGT1b signal relative to Coomassie blue staining control in (G) (mean ± SEM of 6 independent biological repeats, p-values are shown, *t*-test). (I) MPK3 and MPK6 are required for *Pto* AvrRpt2-induced SGT1 phosphorylation. Twelve-d-old Col-0, MPK3SR, and MPK6SR plants grown in liquid medium were first treated with 2 μM NA-PP1 or DMSO (mock) for 1 h. Then the plants were immersed in *Pto* AvrRpt2 (OD600 = 0.02) for the indicated periods of time. In (I), a custom antibody was used to detect the phosphorylated peptide containing AtSGT1 T346. The induction of MAPK activation was determined by western blot using anti-pMAPK antibody. (J) Quantification of pSGT1b signal relative to Coomassie blue staining control in (I) (mean ± SEM of 3 independent biological repeats, p-values, compared to their control at 0 hpi, are shown, *t*-test). These experiments were repeated at least 3 times with similar results. In western blot assays, protein marker sizes are provided for reference. Blots were stained with Coomassie Brilliant Blue (CBB) to verify equal loading.

presence of endogenous NbSGT1. However, tissues expressing both the T346D mutant (2D) or the double S271D/T346D (DD) mutant, but not the single S271D mutant, showed an enhancement of cell death triggered by RPS2 expression, monitored as ion leakage (Fig 5A), which correlated with the appearance of tissue necrosis (Fig 5B and 5C, S10C Fig). Similarly, co-expression with the 2D mutant enhanced cell death triggered by RipE1 and Avr3a/R3a (S10D and S10E Fig), but not BAX (S10F Fig). This suggests that T346 phosphorylation contributes to the robust activation of SGT1-dependent ETI responses, including that triggered by the *R. solanacearum* T3E RipE1.

Given the importance of protein-protein interactions in the SGT1 complex, we hypothesized that phosphorylation in S271/T346 may be important to regulate the interaction of SGT1 with NLRs. Interestingly, the T346A mutation did not affect the interaction between AtSGT1b and RPS2 (Fig 5D and 5E, S10G Fig); however, surprisingly, the T346D mutation inhibited this interaction (Fig 5D and 5E, S10G Fig). This, together with the observation that the T346D mutation causes an enhancement of RPS2-triggered responses, may suggest a scenario where phosphorylation contributes to the release of RPS2 from the SGT1 complex to activate immune responses. Neither the T346A nor T346D mutation affected the interaction between SGT1 and RipAC (S10H and S10I Fig).

## RipAC interferes with the MAPK-SGT1 interaction to suppress SGT1 phosphorylation

During our analysis of SGT1 phosphorylation, we observed an apparent reduction in the number of SGT1 phosphorylated peptides in the presence of RipAC (S7A Fig). Since this assay does not allow a comparative quantitative assessment of SGT1 phosphorylation, we took advantage of the sensitivity of our anti-pSGT1 antibody to determine the impact of RipAC on the phosphorylation of AtSGT1 T346. Interestingly, Arabidopsis transgenic lines expressing RipAC showed reduced phosphorylation of T346 in endogenous SGT1 in basal growth conditions (Fig 6A and 6B). Such reduction of T346 phosphorylation was also observed when Arabidopsis protoplasts were transfected with RipAC in the presence of AvrRpt2 (S11A and S11B Fig). We then generated Arabidopsis transgenic lines expressing RipAC and dexamethasone-inducible AvrRpt2 by crossing. RipAC completely abolished MAPK activation and T346 phosphorylation induced by the expression of AvrRpt2 (6C and 6D Fig). It is noteworthy that infiltration

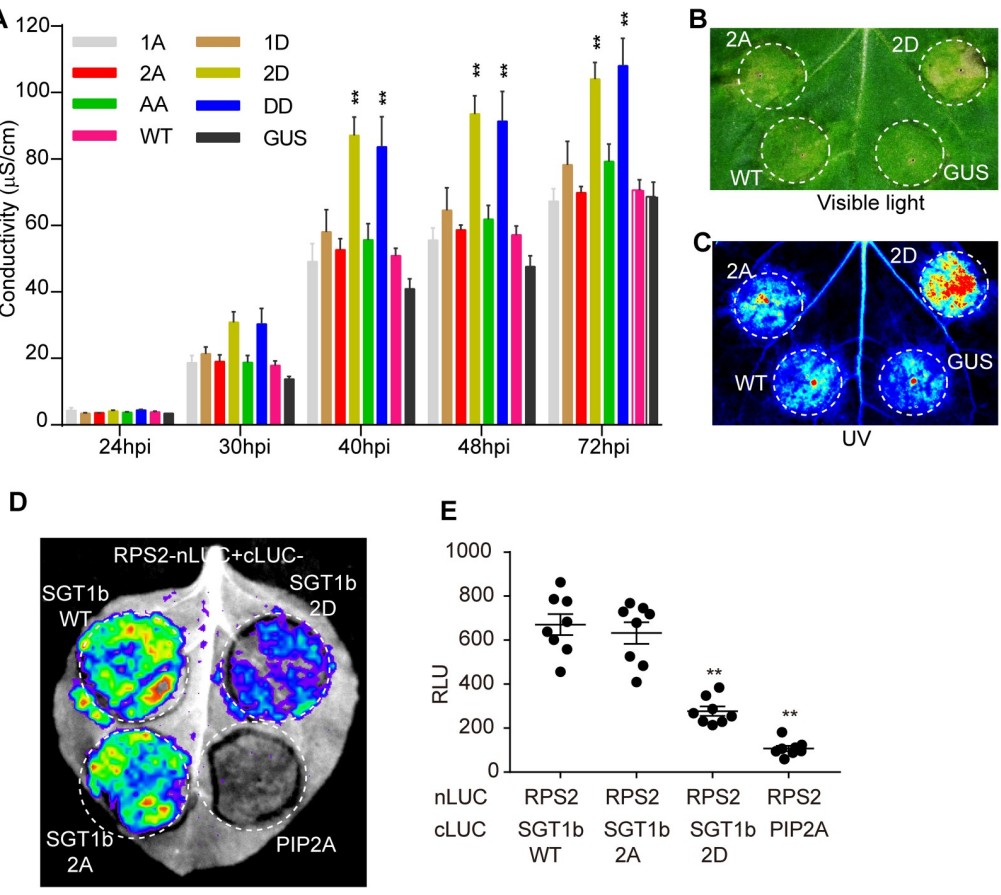

**Fig 5. MAPK-mediated phosphorylation is important for SGT1 function in the activation of ETI.** (A) A phospho-mimic mutation in AtSGT1b T346 (T346D) promotes cell death triggered by RPS2 overexpression in *N. benthamiana*. Agrobacterium expressing AtSGT1b variants or the GUS-FLAG control (OD600 = 0.5) were infiltrated into *N. benthamiana* leaves 1 day before infiltration with Agrobacterium expressing RPS2 (OD600 = 0.15). Leaf discs were taken 21 hpi for conductivity measurements at the indicated time points. The time points in the x-axis are indicated as hpi with Agrobacterium expressing RPS2 (mean ± SEM, n = 4, ** p<0.01, *t*-test, 3 replicates). (B, C) Observation of RPS2-triggered cell death by visible light (B) and UV light (C) in *N. benthamiana*. The Agrobacterium combinations were infiltrated as described in (A), and the cell death phenotype was recorded 4 dpi with Agrobacterium expressing RPS2. (D, E) A phospho-mimic mutation in AtSGT1b T346 (T346D) disrupts RPS2-SGT1b association in *N. benthamiana*. Agrobacterium combinations with different constructs were infiltrated in *N. benthamiana* leaves and luciferase activities were examined in both qualitative (CCD imaging machine, D) and quantitative assays (microplate luminescence reader, E). Quantification of the luciferase signal was performed with microplate luminescence reader (mean ± SEM, n = 8, ** p<0.01, *t*-test, 3 replicates). The nomenclature of the AtSGT1b mutants used is: 1A = S271A, 1D = S271D, 2A = T346A, 2D = T346D, AA = S271AT346A, DD = S271DT346D. The experiments were performed at least 3 times with similar results.

of Arabidopsis leaves with the *R. solanacearum ΔripAC* mutant led to a slight increase in SGT1 phosphorylation, which was not observed upon inoculation with wild-type *R. solanacearum* GMI1000 (6E and 6F Fig). Altogether, these data indicate that RipAC suppresses the phosphorylation of AtSGT1 T346, and provides an additional link between MAPK activation and SGT1 phosphorylation. We considered the possibility that RipAC suppresses SGT1 phosphorylation by interfering with the MAPK-SGT1 interaction. This interference was indeed confirmed by Split-LUC (for MAPK3 and MAPK4) and CoIP assays (for MAPK6), which showed that RipAC inhibits the interaction of SGT1 with MAPKs (Fig 6G–6I, S11C–S11F Fig), suggesting that this could be the mechanism underlying RipAC suppression of SGT1-dependent ETI responses. We did not detect interaction between RipAC and MAPK3/4/6 (S11G and

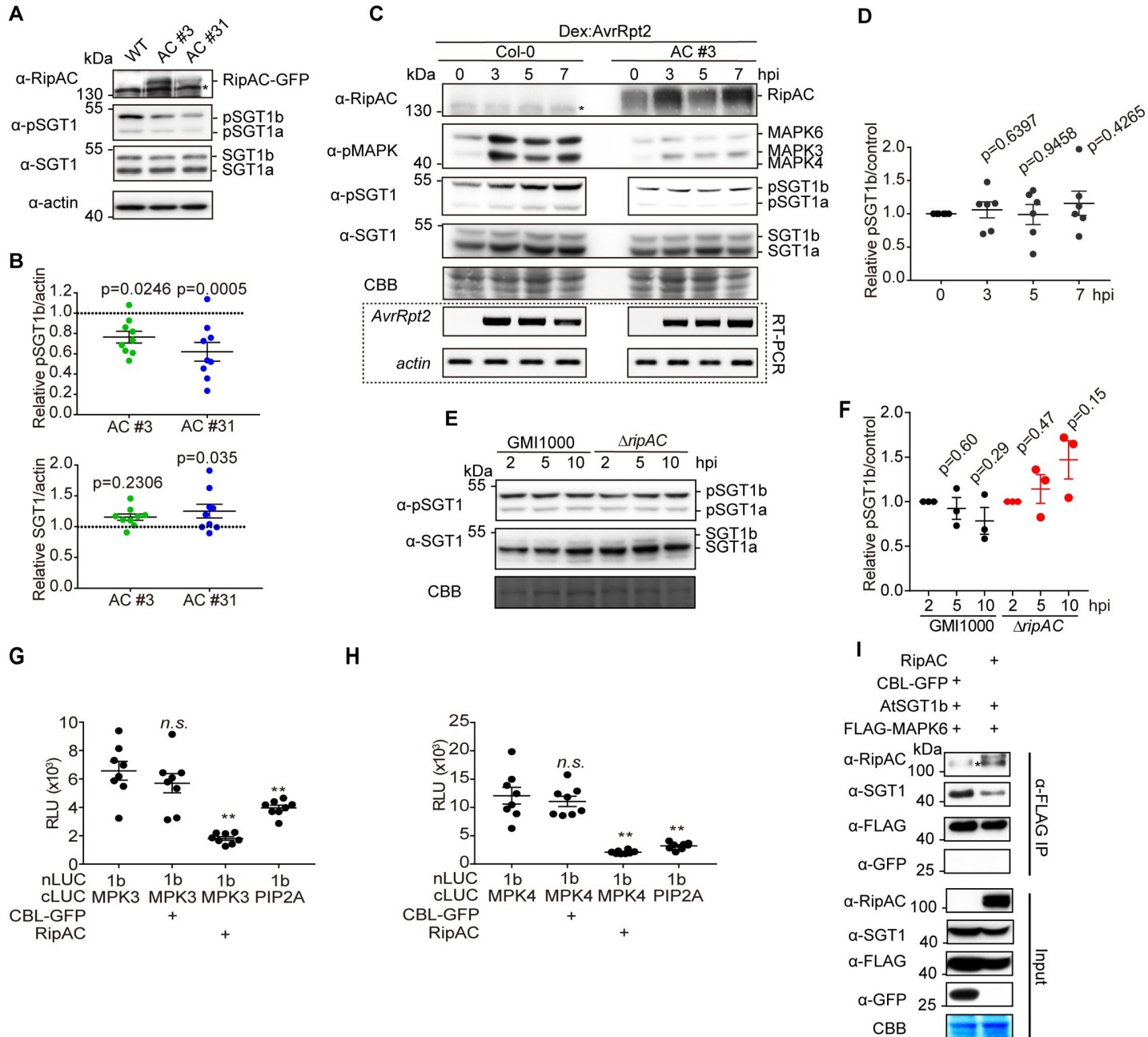

**Fig 6. RipAC interferes with MAPK-SGT1 interaction to suppress SGT1 phosphorylation.** (A) RipAC reduces SGT1 phosphorylation in Arabidopsis. SGT1 phosphorylation was determined in twelve-day-old Col-0 WT and RipAC transgenic Arabidopsis lines (AC #3 and AC #31) by western blot. Custom antibodies were used to detect the accumulation of SGT1 and the presence of a phosphorylated peptide containing AtSGT1 T346. (B) Quantification of SGT1 and pSGT1b signal normalized to actin and relative to Col-0 WT samples in (A) (mean ± SEM, n = 9, p-values are shown, *t*-test). (C) RipAC suppresses ETI-triggered SGT1 phosphorylation in Arabidopsis. ETI triggered MAPK activation and SGT1 phosphorylation were determined in twelve-day-old Dex::*AvrRpt2*/Col-0 or Dex::*AvrRpt2*/ RipAC crossing F1 plants by western blot. Custom antibodies were used to detect the accumulation of SGT1 and the presence of a phosphorylated peptide containing AtSGT1 T346. The induction of AvrRpt2 was determined by RT-PCR. (D) Quantification of pSGT1b signal relative to coomassie blue staining control in (C) (mean ± SEM of 6 independent biological replicates, p-values are shown, *t*-test). (E) Inoculation of *ΔripAC* mutant on Col-0 wild-type plants showing increased SGT1 phosphorylation. Four to five-week-old Col-0 WT plant leaves were infiltrated with GMI1000 WT or *ΔripAC* mutant strain (OD600 = 0.02) and the samples were taken at the indicated time points. Custom antibodies were used to detect the phosphorylated peptide containing AtSGT1 T346 and endogenous SGT1. (F) Quantification of pSGT1b signal relative to Coomassie blue staining control in (E) (mean ± SEM of 3 independent biological repeats, p-values, compared to their control at 2 hpi, are shown, *t*-test). (G, H) Competitive Split-LUC assays showing that RipAC interferes with the interaction between MAPK3 (G) / MAPK4 (H) and AtSGT1b in *Nicotiana benthamiana*. In quantitative assays, the graphs show the mean value ± SEM (n = 8, ** p<0.01, *t*-test). (I) Competitive CoIP showing that RipAC interferes with the interaction between MAPK6 and AtSGT1b in *N. benthamiana*. Anti-FLAG beads were used to IP MAPK6. In all the competitive interaction assays, in addition to the interaction pair, RipAC or CBL-GFP (as negative control) were expressed to determine interference. In all the blots, asterisks indicate non-specific bands. These

experiments were repeated at least 3 times with similar results. In western blot assays, protein marker sizes are provided for reference. Blots were stained with Coomassie Brilliant Blue (CBB) to verify equal loading.

S11H Fig), supporting the idea that RipAC inhibition of MAPK phosphorylation in the context of ETI (Figs 3C–3F and 6C) occurs as a consequence of the attenuation of SGT1 activity and NLR activation, and not by targeting MAPKs directly.

## SGT1 phosphorylation is associated to RipAC virulence activity

Surprisingly, although previous studies found no differences in the susceptibility of *Atsgt1* single mutants to *Pto* DC3000 [46, 47], we found that both *Atsgt1a* and *Atsgt1b* mutants were more susceptible to *R. solanacearum* than their respective Ws-0 and La-*er* wild-type backgrounds, respectively (Fig 7A–7D, and S12A and S12B Fig). In order to determine the biological relevance of SGT1 phosphorylation for RipAC virulence activity during *R. solanacearum* infection, given the impossibility to isolate *Atsgt1a/b* double mutants, we generated Arabidopsis transgenic lines expressing, from a *35S* promoter, AtSGT1b WT, AtSGT1b 2A, or AtSGT1b 2D, abolishing or mimicking phosphorylation in T346, respectively. Overexpression of AtSGT1b WT significantly enhanced resistance against the *ΔripAC* mutant (Fig 7E and 7F, S12C Fig). Overexpression of AtSGT1b 2A partially rescued the virulence attenuation of the *ΔripAC* mutant (Fig 7E and 7F, S12C Fig), suggesting that it may exert a dominant negative effect over the endogenous SGT1, and indicating that SGT1 phosphorylation is relevant for the attenuation phenotype in the absence of RipAC. Intriguingly, overexpression of AtSGT1b 2D dramatically enhanced resistance against the *ΔripAC* mutant (Fig 7E and 7F, S12C Fig).

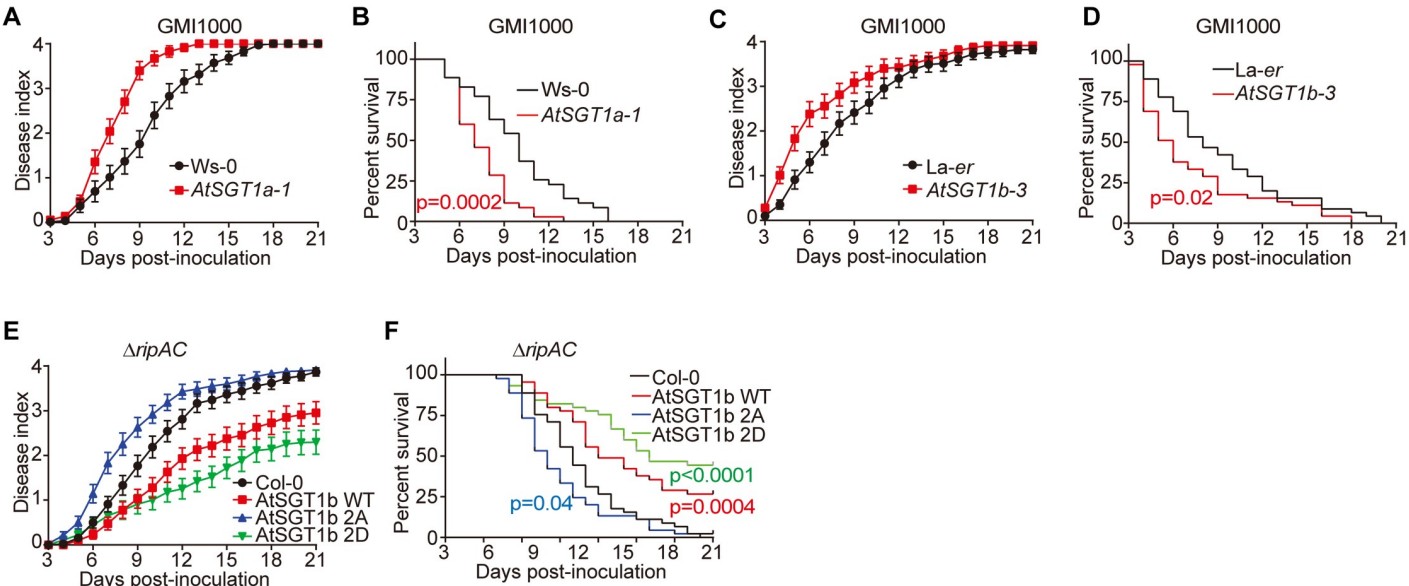

**Fig 7. SGT1 phosphorylation contributes to resistance against *R. solanacearum* in Arabidopsis.** (A-D) Mutation of *AtSGT1* increases the susceptibility to *R. solanacearum* in Arabidopsis. Soil-drenching inoculation assays in different Arabidopsis genotypes (*AtSGT1a-1* and its wild-type control Ws-0, *AtSGT1b-3* and its wild-type control La-*er*) were performed with GMI1000 WT strain. In (A, C) the results are represented as disease progression, showing the average wilting symptoms in a scale from 0 to 4. Values from 3 independent biological repeats were pooled together (mean ± SEM; n = 36 in (A) and n = 48 in (C)). Curves for each replicate are shown in S12A and S12B Fig. (B, D) Survival analysis of the data in (A, C); the disease scoring was transformed into binary data with the following criteria: a disease index lower than 2 was defined as '0', while a disease index equal or higher than 2 was defined as '1' for each specific time point. Statistical analysis was performed using a Log-rank (Mantel-Cox) test (n = 36 in (A) and n = 48 in (C)), and the corresponding p value is shown in the graph with the same colour as each curve. (E, F) Soil-drenching inoculation assays in Arabidopsis transgenic lines overexpressing AtSGT1b variants were performed with GMI1000 *ΔripAC* mutant. In (E) and (F) the analyses were performed the same as in (A) and (B) (n = 45 for each genotype), and the corresponding p value is shown in the graph in (F) with the same colour as each curve. Curves for each replicate are shown in S12C Fig.

## Discussion

Plant invasion by bacterial pathogens leads to the recognition of bacterial elicitors, either PAMPs or effectors, which, in turn, promotes the production and activation of additional immune receptors. Such feedback loop contributes to the robustness of the plant immune system. To achieve a successful infection and establish a compatible interaction, bacteria should evolve to suppress the activation of the immune responses triggered by the recognition of their PAMPs or effectors. Despite the importance of T3Es for the bacterial infection process, single effector knockout mutants often lack virulence phenotypes due to functional redundancy among effectors. This may be particularly important in strains belonging to the *R. solanacearum* species complex, where single strains may secrete more than 70 T3Es [16]. Although such large number of effectors contributes to the infection process, this also poses a risk for *R. solanacearum*, since each effector could potentially be recognized by intracellular NLRs, resulting in ETI. To maintain pathogenicity, bacterial pathogens need to adapt to new recognition events by losing these recognized effectors or evolving additional effectors that suppress ETI. This pathoadaptation becomes evident in the case of RipE1: a *R. solanacearum* GMI1000 derivative strain carrying mutations in *popP1* and *avrA* is pathogenic in *N. benthamiana* [48], despite secreting RipE1, which activates ETI in this plant species [33]. This suggests that *R. solanacearum* has evolved T3Es that suppress immunity triggered by RipE1 (and potentially other T3Es).

In this work, we found that the *R. solanacearum* T3E RipAC interacts with SGT1 (Fig 2), a major regulator of the activity of NLRs in the activation of ETI [49]. SGT1 has been shown to regulate NLR homeostasis, either by contributing to protein stability or by mediating their degradation [47, 50]. RipAC was able to interact with a truncated SGT1 containing only the CS+SGS domains, but not with the CS domain alone, suggesting that RipAC interacts with the SGS domain, or that it requires both CS and SGS domains for interaction. The SGS domain is essential for SGT1 immune functions and mediates SGT1 interaction with the LRRs of NLRs and the co-chaperone HSC70 [24], although its biochemical function remains unclear.

We found that RipAC prevents the interaction between MAPKs and SGT1 in plant cells, leading to a reduced phosphorylation of SGT1 in the SGS domain (Fig 6), suggesting that MAPKs also interact with this domain. Moreover, we found that AtMAPK3 and AtMAPK6 are able to phosphorylate AtSGT1a/b *in vitro*, and that SGT1 phosphorylation *in planta* is compromised by a loss of function of AtMAPK3 and AtMAPK6 (Fig 4). MAPKs have been shown to phosphorylate SGT1 from tobacco and maize in *N. benthamiana* [39, 41], contributing to the hypothesis that MAPK-mediated phosphorylation of SGT1 may be important for its function. This notion is also supported by our data, showing that mutations that mimic phosphorylation in the observed SGT1 phosphosites lead to an enhanced activity of the NLR RPS2 and other SGT1-dependent ETI responses (Fig 5, S10 Fig). Interestingly, the enhanced RPS2 activity caused by the T346D phospho-mimic mutation correlated with an enhanced dissociation between SGT1 (T346D) and RPS2 (Fig 5), suggesting that MAPK-mediated phosphorylation of SGT1 may contribute to NLR activation through the modulation of SGT1 interactions with NLRs. Considering that NLR activation subsequently enhances MAPK activation, this regulatory mechanism could represent a positive feedback loop between the activation of NLRs and MAPKs, mediated by the phosphorylation of SGT1. This notion is supported by the fact that RipAC, which inhibits SGT1 phosphorylation, does not interact with MAPKs (S11G and S11H Fig), but inhibits the MAPK phosphorylation triggered upon activation of NLRs (Figs 3C–3F and 6). Intriguingly, although RipAC suppresses SGT1-dependent immune responses, it does not seem to affect the accumulation of SGT1 or NLRs (in this case, RPS2), which resembles the phenotype observed upon overexpression of the SGS-interacting protein

HSC70 [24]. Moreover, a mutant including a premature stop codon at P347 in AtSGT1b (disrupting the MAPK phosphorylation motif that includes T346) does not affect SGT1 accumulation, but abolishes the immune-associated functions of SGT1 [49], highlighting the importance of the MAPK phosphorylation motif for SGT1 function. Altogether, our data suggest that, by associating with the SGS domain, RipAC compromises SGT1 interaction with MAPKs and the subsequent phosphorylation of specific sites in the SGS domain, compromising the activation of immunity without altering the accumulation of associated proteins.

We have recently shown that RipE1-triggered immunity requires SGT1 [33]. In this work, we show that SGT1 phosphorylation contributes to RipE1-triggered cell death (S10 Fig), and, accordingly, RipAC inhibits RipE1-triggered cell death (Fig 3). Altogether, this suggests that RipAC may contribute to *R. solanacearum* infection process by inhibiting SGT1-dependent immunity triggered by RipE1 and potentially other T3Es. Before, a *ΔripAC* mutant strain was reported to show reduced fitness, compared to GMI1000, in eggplant leaves [51]. Here we show that an equivalent mutant has impaired virulence in Arabidopsis and tomato, and Arabidopsis transgenic plants expressing RipAC become more susceptible to *R. solanacearum* infection (Fig 1), indicating that RipAC plays a significant role in the *R. solanacearum* infection process. Furthermore, we show that the attenuation displayed by the *ΔripAC* mutant is partially rescued by the overexpression of AtSGT1b T346A (2A) (Fig 7). While these results suggest that SGT1 phosphorylation is an important target underlying RipAC contribution to virulence, they do not exclude the possibility that other targets are associated to additional virulence activities of RipAC.

Although the contribution of SGT1 to the establishment of ETI is well known, genetic analysis of its contribution to disease resistance is challenging, due to the deleterious effects caused by the abolishment of SGT1 expression [26]. In this work we found that mutation of *AtSGT1a* or *AtSGT1b* in Arabidopsis enhances susceptibility to *R. solanacearum* (Fig 7). Interestingly, overexpression of AtSGT1b enhanced plant resistance against the *ΔripAC* mutant (Fig 7), indicating that the contribution of AtSGT1b overexpression is particularly evident in the absence of RipAC. It is noteworthy that overexpression of AtSGT1b T346D (2D) strongly enhanced plant resistance against the *ΔripAC* mutant (Fig 7), suggesting that, besides the effect of RipAC on SGT1 phosphorylation, there is an additive effect with other potential virulence activities of RipAC. Altogether, these data highlight the relevance of the SGT1 immunity node and SGT1 phosphorylation in the context of *R. solanacearum* infection.

The T3E AvrBsT from *Xanthomonas campestris* also interacts with SGT1 from pepper, and interferes with its phosphorylation by the PIK1 kinase in two serine residues (in the CS domain and variable region) different from the ones identified in our study [40]. However, in that case, AvrBsT triggers SGT1-mediated immunity, and the identified SGT1 phosphorylation sites are required for the activation of AvrBsT-triggered HR [40]. Similarly, a secreted effector from the fungal pathogen *Ustilago maydis* also interferes with the MAPK-mediated phosphorylation of SGT1 from maize in a monocot-specific phosphorylation site between TPR and CS domains, contributing to the development of disease-associated tumors [41]. This demonstrates that pathogens from different kingdoms and with different lifestyles have evolved to target different functions of SGT1 as a key regulator of disease resistance. However, some of these effectors are indeed perceived by the plant immune system, potentially as a consequence of the 'guarding' of SGT1 as an important regulator of plant immunity. On the contrary, RipAC seems to manipulate SGT1 for virulence purposes without being detected by the plant immune system, suggesting that these host plants have not yet evolved a perception system for this targeting. Together with other reports, our discovery of the molecular mechanisms underlying the RipAC-mediated suppression of SGT1 function may contribute to the

engineering of functional SGT1 complexes that abolish bacterial manipulation or respond to it by activating immunity.

## Materials and methods

The biological and chemical resources used in this study are summarized in the S1 Table. Primers used in this study are summarized in the S2 Table.

### Plant materials

Regarding *Arabidopsis thaliana*, Dexamethasone (Dex)-inducible *Arabidopsis thaliana GVG*: *AvrRpt2* (*Dex:AvrRpt2*) [52], *Dex:MKK5DD* [43], MPK3SR28 (*mpk3mpk6* P$_{MPK3}$:*MPK3* TG, line #28) and MPK6SR58 (*mpk3mpk6* P$_{MPK6}$:*MPK6* YG, line #58) lines [44, 45] in ecotype Col-0, *AtSGT1a-1* (Ws-0) [50], and *AtSGT1b-3* (La-*er*) [46] with their respective wild-type (WT) controls, have been described previously. *GVG:AvrRpt2* (*Dex:AvrRpt2*) was used to cross with either RipAC line #3 or Col-0, and the resulting F1 progenies were used for experiments. In the experiments with Arabidopsis seedlings in 1/2 MS media, the seedlings were kept on 1/2 MS plates in a growth chamber (22˚C, 16 h light/8 h dark, 100–150 mE m$^{-2}$ s$^{-1}$) for germination and growth for 5 days, then transferred to 1/2 MS liquid culture for additional 7 days. For *Pseudomonas syringae* and *Ralstonia solanacearum* infection assays, Arabidopsis plants were cultivated in jiffy pots (Jiffy International, Kristiansand, Norway) in a short day chamber (22˚C, 10 h light/14 h dark photoperiod, 100–150 mE m$^{-2}$ s$^{-1}$, 65% humidity) for 4–5 weeks. After soil drenching inoculation, the plants were kept in a growth chamber under the following conditions: 75% humidity, 12 h light, 130 mE m$^{-2}$ s$^{-1}$, 27˚C, and 12 h darkness, 26˚C for disease symptom scoring.

   *Nicotiana benthamiana* plants were grown at 22˚C in a walk-in chamber under 16 h light/8 h dark cycle and a light intensity of 100–150 mE m$^{-2}$ s$^{-1}$.

   Tomato plants (*Solanum lycopersicum* cv. Moneymaker) were cultivated in jiffy pots (Jiffy International, Kristiansand, Norway) in growth chambers under controlled conditions (25˚C, 16 h light/8 h dark photoperiod, 130 mE m$^{-2}$ s$^{-1}$, 65% humidity) for 4 weeks. After soil drenching inoculation, the plants were kept in a growth chamber under the following conditions: 75% humidity, 12 h light, 130 mE m$^{-2}$ s$^{-1}$, 27˚C, and 12 h darkness, 26˚C for disease symptom scoring.

### Bacterial strains

*Pseudomonas syringae* pv. *tomato* (*Pto*) strains, including *Pto* containing an empty vector (EV), or a vector expressing AvrRpm1, AvrRpt2, or AvrRps4 were cultured overnight at 28˚C in LB medium containing 25 μg mL$^{-1}$ rifampicin and 25 μg mL$^{-1}$ kanamycin.

   *Ralstonia solanacearum* strains, including the phylotype I reference strain GMI1000, GMI1000 Δ*ripAC* mutant, and GMI1000 *ripAC*$^+$ complementation strains, were cultured overnight at 28˚C in complete BG liquid medium [53].

   *Agrobacterium tumefaciens* strain GV3101 with different constructs was grown at 28˚C on LB agar media with appropriate antibiotics. The concentration for each antibiotic is: 25 μg mL$^{-1}$ rifampicin, 50 μg mL$^{-1}$ gentamicin, 50 μg mL$^{-1}$ kanamycin, 50 μg mL$^{-1}$ spectinomycin.

### Generation of plasmid constructs, transgenic plants, and *R. solanacearum* mutant strains

The primers used to generate constructs in this work are presented in S2 Table. To generate the RipAC-GFP construct, the RipAC coding region (RSp0875) in pDONR207 (a gift from

Anne-Claire Cazale and Nemo Peeters, LIPM, Toulouse, France) was sub-cloned into pGWB505 via LR reaction, resulting in pGWB505-RipAC, in which the expression of the *RipAC-GFP* fusion is driven by a *CaMV 35S* promoter. The pGWB502-AtSGT1b variants (WT, 2A, 2D) and pGWB505-RipAC recombinant constructs were transformed into *A. tumefaciens* GV3101 and was then transformed in Arabidopsis Col-0 wild-type (WT) through the floral dipping method [54]. Transgenic Arabidopsis plants were selected using hygromycin (50 μg mL$^{-1}$) and were further confirmed by western blot using an anti-RipAC custom antibody. All the experiments using RipAC transgenic Arabidopsis were performed using two independent T4 homozygous lines, which were germinated and grown in the same conditions as wild-type plants.

To generate the *R. solanacearum* Δ*ripAC* mutant, the *RipAC* coding region was replaced by gentamicin resistant gene using homologous recombination method [55]. The *RipAC* flanking regions, left border (LB) and right border (RB), were amplified by PCR and were recombined in the pEASYBLUNT vector, while the gentamicin resistance gene was inserted between the LB and RB by *Eco*R I digestion and T4 ligation. The resulting plasmid pEASYBLUNT-LB-Gm-RB was introduced into *R. solanacearum* GMI1000 WT strain by natural transformation [56]. The Δ*ripAC* mutant was selected using gentamicin (10 μg mL$^{-1}$) and PCR using RipAC coding region specific primers (S2 Table). To complement *RipAC* in the Δ*ripAC* mutant, the *RipAC* promoter (423 bp upstream of ATG of the *RipABC* operon) [17] was cloned into pRCT-GWY [57], after which the *RipAC* coding region was introduced by LR reaction to result in a pRCT-pRipAC-RipAC construct. The integrative pRCT-pRipAC-RipAC plasmid, triggering the expression of RipAC under the control of the native *RipABC* operon promoter, was transformed into *R. solanacearum* Δ*ripAC* mutant strain by natural transformation. The complementation strain *ripAC*$^{+}$ was selected using Tetracycline (10 μg mL$^{-1}$) and PCR using the RipAC coding region specific primers (S2 Table).

## Pathogen inoculation assays

For *R. solanacearum* in soil inoculation, 12–15 four-to-five-week old Arabidopsis plants per genotype or 12 tomato plants for each bacterial strain (grown in Jiffy pots) were inoculated by soil drenching with a bacterial suspension containing 10$^{8}$ colony-forming units per mL (CFU mL$^{-1}$). 300 mL of inoculum of each strain was used to soak each treatment. After 20-minute incubation with the bacterial inoculum, plants were transferred from the bacterial solution to a bed of potting mixture soil in a new tray [58]. Scoring of visual disease symptoms on the basis of a scale ranging from '0' (no symptoms) to '4' (complete wilting) was performed as previously described [58]. To perform the survival analysis, the disease scoring was transformed into binary data with the criteria: the disease index lower than 2 was defined as '0', while the disease index equal or higher than 2 was defined as '1' in terms of the corresponding time (days post-inoculation, dpi) [59]. For stem injection assays, 5 μL of a 10$^{6}$ CFU mL$^{-1}$ bacterial suspension was injected into the stems of 4-week-old tomato plants and 2.5 μL xylem sap was collected from each plant for bacterial number quantification 3 dpi. Injections were performed 2 cm below the cotyledon emerging site in the stem, while the samples were taken at the cotyledon emerging site.

To infiltrate *R. solanacearum* GMI1000 or Δ*ripAC* mutant strains, 4-5-week-old short day grown Arabidopsis Col-0 WT plants were moved to a *Ralstonia* inoculation chamber (with the environmental conditions described above) 2 days before pathogen inoculation. *Ralstonia* strains were resuspended in water at 1x10$^{7}$ CFU mL$^{-1}$. The bacterial suspensions were then infiltrated into Arabidopsis leaves and the samples were taken at 2, 5, and 10 hours post-inoculation (hpi). The collected samples were subjected to protein extraction followed by western blots.

For *Pto* inoculation, different *Pto* strains were resuspended in water at $1x10^5$ CFU mL$^{-1}$. The bacterial suspensions were then infiltrated into 4-5-week-old Arabidopsis leaves. Bacterial numbers were determined 3 dpi.

To inoculate *Pto* AvrRpt2 on *MPK3SR*, *MPK6SR*, or Col-0 WT plants, 12-day old MS media-grown Arabidopsis seedlings were treated with 2 μM NA-PP1 for 1h to block the MAPK activity as previously described [60]. Then the seedlings were submerged in buffer containing *Pto* AvrRpt2 at $1x10^7$ CFU mL$^{-1}$ under growth light, and the samples were taken 0, 6, and 12 hpi. DMSO was used to treat Col-0 WT plants as solvent control. The collected samples were subjected to protein extraction followed by western blots.

## Transient gene expression and ion leakage measurements

For protein accumulation, microsome fractionation, confocal microscopic observation, split-luciferase complementation (Split-LUC), and co-immunoprecipitation assays, *A. tumefaciens* GV3101 carrying different constructs were infiltrated into leaves of 5-week-old *N. benthamiana*. The OD600 used was 0.5 for each strain in all the assays, except for Split-LUC assays, for which we used OD600 = 0.2. To prepare the inoculum, *A. tumefaciens* was incubated in the infiltration buffer (10 mM MgCl$_2$, 10 mM MES pH 5.6, and 150 μM acetosyringone) for 2 h. For effector-triggered immunity suppression assay, *A. tumefaciens* expressing RipAC-GFP or GFP were infiltrated into *N. benthamiana* leaves with an OD600 = 0.5, one day after which *A. tumefaciens* expressing the cell death-inducing agents were infiltrated at the same spots with an OD600 = 0.15. Pictures were taken 5-d-post expression of the cell death-inducing agent. Cell death was observed using a BIO-RAD GelDoc XR$^+$ with Image Lab software.

For ion leakage measurement assays, the AtSGT1b variants were infiltrated at an OD600 of 0.5 and the cell death-inducing agents: 35S-RPS2-HA, Avr3a/R3a pair, RipE1, or BAX, was infiltrated at an OD600 of 0.15 in the same infiltrated spots one-day-post infiltration with *A. tumefaciens* expressing the AtSGT1b variants. Three leaf discs (diameter = 10mm) were collected per infiltration site into 4 mL ddH$_2$O in 12-well plate at 21-h-post infiltration with *A. tumefaciens* expressing cell death-inducing agents, then were placed on the bench top for 1 hour to remove the ion leakage caused by the wounding. Leaf discs were carefully transferred to fresh plates with ddH$_2$O. Ion leakage was determined with ORION STAR A212 conductivity meter (Thermo Scientific) at the indicated time points. In the graph, 24 hpi indicates 24-hours post infiltration with *A. tumefaciens* expressing cell death-inducing agents.

## Protein extraction, microsome fractionation, and immunoblot analysis

To extract protein samples, 12-day-old Arabidopsis seedlings and leaf discs (diameter = 18mm) from *N. benthamiana* were frozen in liquid nitrogen and ground with a Tissue Lyser (QIAGEN, Hilden, Nordrhein-Westfalen, Germany). Samples were subsequently homogenized in protein extraction buffer (100 mM Tris (pH 8), 150 mM NaCl, 10% Glycerol, 1% IGEPAL, 5 mM EDTA, 5 mM DTT, 1% Protease inhibitor cocktail, 2 mM PMSF, 10 mM sodium molybdate, 10 mM sodium fluoride, 2 mM sodium orthovanadate). To isolate the microsome fraction, proteins were extracted in buffer H (250 mM sucrose, 50 mM N-(2-hydroxyethyl) piperazine-N'-(2-ethanesulfonic acid) (HEPES)-KOH (pH 7.5), 5% (v/v) glycerol, 50 mM sodium pyrophosphate decahydrate, 1 mM sodium molybdate dihydrate, 25 mM sodium fluoride, 10 mM EDTA, 0.5% (w/v) polyvinylpyrrolidone (PVP-10) with a final concentration of 3 mM DTT and 1% protease inhibitor cocktail) and subjected to microsomal protein enrichment using differential centrifugation and Brij-58 treatment as previously described [61]. The resulting protein samples were boiled at 70˚C for 10 minutes in Laemmli buffer and loaded in SDS-PAGE acrylamide gels for western blot. All the immunoblots were

analyzed using appropriate antibodies as indicated in the figures. Molecular weight (kDa) marker bands are indicated for reference.

## Generation of custom antibodies

To raise anti-RipAC antibodies, purified recombinant GST-RipAC$_{241-540aa}$ expressed in *E. coli* was used as antigen [62]. The polyclonal antiserum from immunized rabbits was purified by affinity chromatography using recombinant GST-RipAC$_{241-540aa}$, and the eluate was used as anti-RipAC antibody (Abclonal Co., Wuhan, China). The anti-RipAC antibody specificity was determined by using RipAC-GFP transient expression in *N. benthamiana*. The SGT1 phosphosite-specific antibodies (anti-AtSGT1b pT346) were generated by Abclonal Co. Briefly, the synthesized phospho-peptide [ES(T-p)PPDGME-C] was conjugated to keyhole limpet hemocyanin (KLH) carrier to immunize rabbits. The rabbit polyclonal antiserum was purified by affinity chromatography using phospho-peptide and the eluate was cleaned by passing through the column coupled with control synthesized-peptide (ESTPPDGME-C) to remove the non-specific antibodies. To determine the pSGT1 antibody specificity, an *in vitro* ELISA and an *in vivo* Calf Intestinal Alkaline Phosphatase (CIAP) treatment were performed. In the ELISA assay, the phospho-peptide and control peptide were spotted on the nitrocellulose filter membrane and probed with different diluted anti-pSGT1 antibody. In the CIAP treatment, protein samples from two 12-d-old Col-0 seedlings were extracted with 300 μL 1 x NEB CutSmart buffer (50 mM Potassium Acetate, 20 mM Tris-acetate, 10 mM Magnesium Acetate, 100 μg/ml BSA, pH 7.9) with 1% protease inhibitor cocktail. 2 μL CIAP enzyme (New England Lab) was added to a vial of 40 μL protein extract, while another 40 μL protein extract with no CIAP was used as control. The treatment was performed by incubating protein extract at 37˚C for 1h, after which the protein sample was subjected to western blot analysis with different antibodies. The anti-pSGT1 antibody was used to determine the effect of CIAP treatment, while the anti-SGT1 antibody [63] was used to show the endogenous SGT1 abundance.

## MAP kinase activation assays

To measure the MAPK activation triggered by cell death inducers, *N. benthamiana* leaf discs were collected 24-h-post infiltration with *A. tumefaciens* expressing the cell death inducers. To evaluate the MAPK activation in *Dex:AvrRpt2* Arabidopsis, 12-d-old *Dex:AvrRpt2*/Col-0 or *Dex:AvrRpt2*/AC #3 F1 crossing seedlings grown in liquid media were treated with 30 μM Dex for different times as indicated and previously described [64, 65]. To evaluate the MAPK activation in *Dex:MKK5DD* Arabidopsis, 12-d-old *Dex:MKK5DD* seedlings grown in liquid media were treated with 30 μM Dex for different times as indicated. After protein extraction, the protein samples were separated in 10% SDS-PAGE gels and the western blots were probed with anti pMAPK antibodies to determine MAPK activation as previously described [66]. Detection of SGT1 accumulation and SGT1 phosphorylation were also performed with the same samples using custom anti-SGT1 [63] and anti-pSGT1 antibodies. The induction of AvrRpt2 was determined by RT-PCR. Blots were stained with Coomassie Brilliant Blue (CBB) or were probed with anti-actin to verify equal loading.

## Large-scale immunoprecipitation and LC-MS/MS analysis

Large-scale immunoprecipitation assays for LC-MS/MS analysis were performed as previously described with several modifications [67]. Two grams of *N. benthamiana* leaf tissues were collected at 2 days after infiltration with *A. tumefaciens* and frozen in liquid nitrogen. The protein extracts were homogenized in protein extraction buffer (100 mM Tris-HCl pH8, 150 mM NaCl, 10% glycerol, 5 mM EDTA, 10 mM DTT, 2 mM PMSF, 10 mM NaF, 10 mM Na$_2$MoO$_4$,

2 mM NaVO$_3$, 1%(v/v) NP-40, 1%(v/v) plant protease inhibitor cocktail (Sigma). The protein extracts were cleaned by 2 rounds of 10 min x 15, 000 g centrifugation to remove the tissue debris. 30 μL GFP-trap beads (ChromoTek, Germany) or ANTI-FLAG M2 Affinity Agarose Gel (Sigma) was added to the clean protein extract and incubated for one hour at 4˚C with an end-to-end slow but constant rotation. GFP-Trap beads or ANTI-FLAG beads were washed four times with 1 mL cold wash buffer (100 mM Tris-HCl pH 8, 150 mM NaCl, 10% glycerol, 2 mM DTT, 10 mM NaF, 10 mM Na$_2$MoO$_4$, 2 mM NaVO$_3$, 1%(v/v) plant protease inhibitor cocktail (Sigma), 0.5%(v/v) NP-40 for GFP-Trap beads, no NP-40 for anti-FLAG M2 beads) and the washed beads were subjected to Mass Spectrometric analysis as previously described to identify interacting proteins [66] and phosphorylated peptides [68].

## Co-immunoprecipitation

One gram of *N. benthamiana* leaf tissues was collected at 2 days after infiltration with *A. tumefaciens* and frozen in liquid nitrogen. Total proteins were extracted as indicated above and immunoprecipitation was performed with 15 μL of GFP-trap beads (ChromoTek, Germany) or ANTI-FLAG M2 Affinity Agarose Gel (Sigma) as described above. Beads were washed 4 times with wash buffer with different detergent concentrations. To specify, in Fig 2A and 2D, there is no detergent in the wash buffer, while in Figs 4A and 6I, the concentration for detergent in wash buffer is 0.2%. The proteins were stripped from the beads by boiling in 40 μL Laemmli buffer for 10 minutes at 70˚C. The immunoprecipitated proteins were separated on SDS-PAGE gels for western blot analysis with the indicated antibodies. Blots were stained with CBB to verify equal loading.

## Split-LUC assays

Split-LUC assays were performed as previously described [69, 70]. Briefly, *A. tumefaciens* strains containing the desired plasmids were hand-infiltrated into *N. benthamiana* leaves. Split-LUC assays were performed both qualitatively and quantitatively after 2 dpi or 1dpi, respectively, in the RPS2 expression assays. For the CCD imaging, the leaves were infiltrated with 0.5 mM luciferin in water and kept in the dark for 5 min before CCD imaging. The images were taken with either Lumazone 1300B (Scientific Instrument, West Palm Beach, FL, US) or NightShade LB 985 (Berthold, Bad Wildbad, Germany). To quantify the luciferase signal, leaf discs (diameter = 4 mm) were collected into a 96-well microplate (PerkinElmer, Waltham, MA, US) with 100 μL H$_2$O. Then the leaf discs were incubated with 100 μL water containing 0.5 mM luciferin in a 96-well plate wrapped with foil paper to remove the background luminescence for 5 min, and the luminescence was recorded with a Microplate luminescence reader (Varioskan flash, Thermo Scientific, USA). Each data point contains at least eight replicates. The protein accumulation was determined by immunoblot as described above.

## Sequence analysis

The SGT1 protein sequences were retrieved from Phytozome (https://phytozome.jgi.doe.gov/pz/portal.html) and sequence alignments were generated using Clustal Omega (https://www.ebi.ac.uk/Tools/msa/clustalo/) with default settings. The consensus MAPK-mediated phosphorylation sites were generated using WebLogo (https://weblogo.berkeley.edu).

## Confocal microscopy

To determine the protein subcellular localization, bimolecular fluorescence complementation (BiFC), and protein co-localization in *N. benthamiana*, leaf discs were collected 2 dpi and

observed using a Leica TCS SP8 (Leica, Mannheim, Germany) confocal microscope with the following excitation wavelengths: GFP, 488nm; YFP, 514nm; RFP, 561 nm [71].

## Protoplast assays

Protoplast transient expression assays were performed as described previously [72, 73]. In brief, 200 μL of protoplasts ($2 \times 10^6$ cells) were mixed with plasmids (20 μg for each construct) with a 1:1 plasmid ratio for every combination. The protoplasts were incubated at room temperature overnight and were harvested by centrifugation at 100 g for 2 min. After removing the supernatant, 90 μL protein extraction buffer was added to the protoplasts. After boiling in the Laemmli buffer, the protein samples were subjected to western blot analysis.

## Protein expression in *E. coli* and *in vitro* pull-down assays

The fragment encoding full-length RipAC was amplified by PCR and inserted into His-SUMO vector [74] by in-fusion method. The plasmids to express recombinant GST-AtSGT1a, GST-AtSGT1b, and GST-GUS in pGEX-6p-1 [49], His-MPK3, His-MPK6, and His-MKK5DD in pET28a [75] were described previously. The resulting plasmids were transformed into *Escherichia coli* (*E. coli*) BL21 (DE3). The resulting strains were grown in LB medium containing appropriate antibiotics (50 μg mL$^{-1}$ kanamycin for His-SUMO-RipAC, 50 μg mL$^{-1}$ ampicillin for pGEX-6p-1 based constructs) and expression of recombinant proteins was induced by the addition of 0.5 mM isopropyl-β-D-thiogalactopyranoside (IPTG) at 16 ˚C overnight. Cells were then collected and subjected to protein purification using either Glutathione Sepharose 4 Fast Flow (GE Healthcare) for GST purification or Ni-NTA His Bind Resin (Novagen). The buffers used for protein purification are: GST purification (wash buffer: 140 mM NaCl, 2.7 mM KCl, 10 mM $Na_2HPO_4$, 1.8mM $KH_2PO_4$, 1% Triton X-100, pH 8.0; elution buffer: 10 mM reduced Glutathione, 50 mM Tris-HCl, pH8.0); His purification (wash buffer: 20 mM Tris-HCl, 20 mM imidazole, 500 mM NaCl, pH 8.0; elution buffer: 20 mM Tris-HCl, 500 mM imidazole, 500 mM NaCl, pH 8.0).

For *in vitro* pull down assays, after sonication and centrifugation, the GST fusion proteins were enriched from supernatant with Glutathione Sepharose 4 Fast Flow (GE Healthcare). The beads bound with GST-AtSGT1a/AtSGT1b and GST-GUS were washed with GST wash buffer for GST pull-down assays. Fifty microliter aliquots of glutathione-agarose beads were incubated with 4 μg of recombinant His-SUMO-RipAC protein at 4 ˚C in a final 300 μL reaction system with GST purification wash buffer on an earthquake shaker for 1h. The beads were washed four times with 50 mM Tris-HCl, pH 7.4, 100 mM NaCl, 1 mM EDTA, 0.5% IGEPAL by centrifugation at 4 ˚C and boiled with 1xLaemmli buffer for 10 min. The eluted proteins were separated on a 10% SDS-PAGE gel and subjected to immunoblot analysis using the antibody against RipAC antibody and the blots were also visualized with Coomassie brilliant blue staining.

## *In vitro* phosphorylation assays

*In vitro* phosphorylation assays were performed as previously described [75]. Activated recombinant MPK3/MPK6 was generated by incubating His-tagged MPK3/MPK6 (3 μg) with recombinant MKK5DD (1 μg) at 27 ˚C for 30 min in 20 μl of reaction buffer (20 mM Tris-HCl, pH 7.5, 10 mM $MgCl_2$, 1 mM DTT, and 50 μM ATP). Then the recombinant AtSGT1a/b (3 μg) was phosphorylated by activated MPK3/MPK6 in the same reaction buffer at 27 ˚C for 30 min. The reaction was stopped by addition of 1xLaemmli buffer with boiling at 70˚C for 10 min. The proteins were separated on a 10% SDS-PAGE gel and subjected to immunoblot

analysis using the antibody using SGT1 [63] and SGT1 phosphor-antibodies, and the blots were also visualized with Coomassie brilliant blue staining.

## Quantification and statistical analysis

Statistical analyses were performed with Prism 7 software (GraphPad). The data are presented as mean ± SEM. The statistical analysis methods are described in the figure legends.

## Supporting information

**S1 Fig. RipAC encodes a leucine rich repeat protein in *Ralstonia solanacearum*.** (A) RipAC is a core effector present in most sequenced *R. solanacearum* strains. The data is retrieved from the Ralsto T3E website (https://iant.toulouse.inra.fr/bacteria/annotation/site/prj/T3Ev3/ ). (B) Phylogenetic analysis of RipAC proteins from different sequenced *R. solanacearum* strains. The phylogenetic tree was generated using the Maximum Likelihood method based on the JTT matrix-based model. The tree is drawn to scale with branch lengths meaning the measurement number of substitutions per site. (C) RipAC encodes a leucine rich repeat protein with no predicted enzymatic domain. The upper panel is the diagram of the RipAC protein with LRR domain; the lower panel is the detailed analysis of the RipAC LRR domain, and the numbers on the left indicate LRR numbers. (D) Diagram of the process of generation of the *R. solanacearum ΔripAC* mutant. The *ΔripAC* mutant was generated by homologous recombination method using a pEASYBLUNT-based plasmid as described in the methods section. (E) Generation of *RipAC* complementation strain in the *ΔripAC* mutant strain. A 423bp DNA fragment upstream of ATG of the *RipABC* operon was amplified and inserted into pRCT plasmid, and the RipAC coding region was shifted into the pRCT plasmid by LR reaction to result in the pRCT-*pRipAC*-*RipAC* expression cassette. The integrative pRCT-pRipAC-RipAC plasmid was mobilized into the *ΔripAC* mutant by natural transformation to result in *ripAC+*. (F) PCR characterization of the presence of the *RipAC* DNA fragment in different strains. A pair of PCR primers was designed to amplify the *RipAC* full-length coding region and the PCR was performed to examine the presence of *RipAC* gene. (G) Bacterial growth in nutrient-rich medium. GMI1000 WT, *ΔripAC*, and *ripAC+* strains were inoculated into the complete BG liquid medium with initial OD600 = 0.005 and the bacterial growth was monitored at the indicated time points measuring OD600 (mean ± SEM, n = 3).
(TIF)

**S2 Fig. *Ralstonia solanacearum* inoculation assays in Arabidopsis and tomato.** (A) Soil-drenching inoculation assays in Arabidopsis were performed with GMI1000 WT, *ΔripAC* mutant, and RipAC complementation (*ripAC+*) strains. Composite data from 3 independent biological repeats (average values are shown in Fig 1A). n = 15 plants per genotype in each repeat. (B) Soil-drenching inoculation assays in tomato were performed with GMI1000 WT, *ΔripAC* mutant, and RipAC complementation (*ripAC+*) strains. Composite data from 3 independent biological repeats (average values are shown in Fig 1C). n = 12 plants per genotype in each repeat.
(TIF)

**S3 Fig. RipAC-GFP transgenic Arabidopsis are more susceptible to *R. solanacearum* GMI1000.** (A) Typical developmental phenotypes of RipAC-GFP transgenic Arabidopsis. AC #3 and AC #31 are two independent transgenic lines (T4 generation). The picture shows 1-month-old Arabidopsis grown in a short-day growth chamber. (B) Western blot shows RipAC-GFP protein accumulation in transgenic Arabidopsis. Samples were taken at 12 days after germination. Blots were probed with antibody Anti-RipAC (1,5000). (C) Soil-drenching

inoculation assays in RipAC-GFP transgenic lines with GMI1000 WT strain. Composite data from 3 independent biological repeats (average values are shown in Fig 1A). n = 15 plants per genotype in each repeat.
(TIF)

**S4 Fig. Subcellular localization of RipAC-GFP.** (A) Subcellular localization of RipAC-GFP in *Nicotiana benthamiana*. RipAC-GFP or free GFP were transiently expressed in *N. benthamiana* leaves using *Agrobacterium tumefaciens*, and the GFP fluorescence signal was observed 48 hpi using confocal microscopy. Scale bar = 50 μm. (B) Microsome fractionation in *N. benthamiana*. Agrobacterium carrying RipAC or CBL-GFP was infiltrated into 5-week-old *N. benthamiana* leaves and samples were taken at 2dpi and then subjected to microsome fractionation. The total protein extraction was separated into the cytosolic fraction and the microsomal fraction using centrifugation as described in the methods section. Protein samples from total extract, cytosolic, and microsome fraction were used for western blot. The plasma-membrane protein H$^+$-ATPase was used as a microsomal protein marker. Western blots from 3 biological replicates are represented.
(TIF)

**S5 Fig. RipAC associates with SGT1s in plant cells.** (A) RipAC-GFP or free GFP were transiently expressed in *Nicotiana benthamiana* leaves. The figure shows the unique NbSGT1 peptides identified exclusively in RipAC-GFP sample upon GFP immunoprecipitation followed by IP-MS/MS analysis. (B) CBL-GFP localizes at plasma membrane in *N. benthamiana*. bar = 75 μm. (C) Co-localization analyses of plasma membrane-associated CBL-GFP protein and SGT1-RFP in *N. benthamiana*. (D) RipAC-GFP co-localizes with SGT1-RFP in *N. benthamiana*. In (B), (C) or (D) CBL-GFP alone, CBL-GFP with SGT1-RFP (AtSGT1a, AtSGT1b, and NbSGT1) or RipAC-GFP with SGT1-RFP (AtSGT1a, AtSGT1b, and NbSGT1) was transiently expressed using Agrobacterium in *N. benthamiana*, and the GFP fluorescence signal was observed 48 hpi using confocal microscopy. In (C) and (D), the images show the fluorescence from GFP, RFP channel, and the merged fluorescence from both channels. The corresponding fluorescence intensity profiles (GFP, green; RFP, red) across the green lines are shown. Scale bar = 50 μm. (E) Western blot showing protein accumulation in Fig 2B. (F) Split-YFP complementation assay to determine direct interaction between RipAC and SGT1 in *N. benthamiana*. The self-association of aquaporin AtPIP2A was used as a positive interaction control, while the RipAC-AtPIP2A combination was used as a negative control. Fluorescence signal was observed 48 hpi using confocal microscopy. Scale bar = 50 μm. (G) Competitive Split-LUC showing that RipAC does not interfere with RPS2-SGT1b association in *N. benthamiana*. (H) Western blot shows the protein accumulation in (G). The RPS2-PIP2A combination was used as negative control. In all the competitive interaction assays, in addition to the interaction pair, RipAC or CBL-GFP (as negative control) were expressed to determine interference. In (G) luciferase activity was determined both qualitatively (CCD camera, higher panel) and quantitatively (microplate luminescence reader, lower panel). All the experiments were performed 3 times with similar results. In western blot assays, protein marker sizes are provided for reference.
(TIF)

**S6 Fig. RipAC inhibits SGT1-dependent immune responses.** (A) Ion leakage assays showing RipAC specifically suppresses RPS2-, but not BAX-, mediated cell death in *Nicotiana benthamiana*. Agrobacterium expressing RipAC or the GFP control (OD600 = 0.5) were infiltrated into *N. benthamiana* leaves 1 day before infiltration with Agrobacterium expressing RPS2 or BAX (OD600 = 0.15). Leaf discs were taken 21 hpi for conductivity measurements at

the indicated time points. The time points in the x-axis are indicated as hpi with Agrobacterium expressing RPS2 or BAX (mean ± SEM, n = 3, 4 replicates). (B) Ion leakage assays showing RipAC suppresses RipE1-mediated cell death in *N. benthamiana*. The ion leakage assays were performed the same as in (A) (mean ± SEM, n = 3, * p<0.05, ** p<0.01, *t*-test, 3 replicates). (C) RPS2-triggered cell death in *N. benthamiana* was monitored by visible light and UV light. (D) Phosphorylation of NbSIPK and NbWIPK was detected using anti-pMAPK antibody in (C).
(TIF)

**S7 Fig. NbSGT1 phosphorylation in plant cells.** (A) NbSGT1 is phosphorylated in *Nicotiana benthamiana*. Agrobacterium containing NbSGT1-FLAG with RipAC or GFP was infiltrated into 4–5 weeks old *N. benthamiana* plants and samples were harvested 48 hpi and were subjected to anti-FLAG IP-MS/MS. The phosphorylation of S282 and S358 is summarized from three biological IP-MS/MS replicates. (B) Representative MS/MS spectra showing phosphorylation of Ser282 and Ser358 in NbSGT1 expressed in *N. benthamiana*.
(TIF)

**S8 Fig. MAPK-mediated phosphorylation of SGT1 is conserved in plant kingdom.** (A) Phosphorylation of S282 and S358 in NbSGT1 are located in the SGS (SGT1-specific) domain. The S358 is predicted to be a MAPK phosphorylation site. At = *Arabidopsis thaliana*; Ca = *Capsicum annuum*; Nb = *Nicotiana benthamiana*; Zm = *Zea mays*; Sc = *Saccharomyces cerevisiae*; Hs = *Homo sapiens*; Ms = *Mus musculus*. (B) MAPK-mediated phosphorylation motif in SGT1 proteins is conserved in the plant kingdom. SGT1 protein sequences from different plant species genome were retrieved from Phytozome (https://phytozome.jgi.doe.gov/pz/portal.html). Then the sequences were aligned with MEGA program and the MAPK-mediated phosphorylation motif "S/TP" were selected for Weblogo.
(TIF)

**S9 Fig. AtSGT1s associate with AtMAPKs.** (A, B) AtSGT1a/b associate with AtMAPK3/4, but not with MAPK6, in Split-LUC assay in *Nicotiana benthamiana*. Agrobacterium combinations with different constructs were infiltrated in *N. benthamiana* leaves and luciferase activities were examined with CCD imaging machine. The MAPK-PIP2A combination was used as negative control. (C) Protein accumulation in (A) and (B). These experiments were repeated at least 3 times with similar results. In western blot assays, protein marker sizes are provided for reference.
(TIF)

**S10 Fig. Phosphorylation of SGT1 T346 contributes to ETI robustness.** (A) Determination of the specificity of the pSGT1 antibody using an *in vitro* ELISA assay. (B) Determination of the specificity of pSGT1 antibody in Arabidopsis using CIAP treatment. Two 12-d-old Col-0 WT plants were used for protein extract and CIAP enzyme treatment (37˚C, 60min) and the treated protein sample was subjected to western blot with the indicated antibodies. (C) A phospho-mimic mutation in AtSGT1b T346 (T346D) promotes cell death triggered by RPS2 overexpression in *N. benthamiana*. The experiments were performed as in Fig 5A and the cell death phenotype was recorded 4 dpi with Agrobacterium expressing RPS2. (D-F) A phospho-mimic mutation in AtSGT1b T346 (T346D) promotes cell death triggered by RipE1, Avr3a/R3a, but not BAX overexpression in *Nicotiana benthamiana*. Agrobacterium expressing AtSGT1b variants or the GUS-FLAG control (OD600 = 0.5) were infiltrated into *N. benthamiana* leaves 1 day before infiltration with Agrobacterium expressing RipE1, Avr3a/R3a, or BAX (OD600 = 0.15). Leaf discs were taken 21 hpi for conductivity measurements at the indicated time points. The time points in the x-axis are indicated as hpi with Agrobacterium expressing

RipE1, Avr3a/R3a, or BAX (mean ± SEM, n = 3, ** p<0.01, *t*-test, 3 replicates). (G) Western blot shows protein accumulation in Fig 5D and 5E. (H, I) RipAC associates similarly with AtSGT1b variants. The Split-LUC assays were done the same as in Fig 2B. The protein accumulation of each construct is shown in (I). These experiments were repeated at least 3 times with similar results. In western blot assays, protein marker sizes are provided for reference. (TIF)

**S11 Fig. RipAC interferes with MAPK-SGT1 interaction to suppress SGT1 phosphorylation.** (A) RipAC suppresses ETI-triggered SGT1 phosphorylation in Arabidopsis protoplasts. Western blots were performed as in Fig 4A using samples from protoplasts 10 hours after transfection with the indicated constructs. (B) Quantification of pSGT1b signal normalized to actin and relative to the control sample transfected with GUS and GFP in (A) (mean ± SEM of 3 independent biological replicates, * p<0.05, ** p<0.01, *t*-test). (C, D) Competitive Split-LUC assays showing that RipAC interferes with the interaction between MAPK3 (C) / MAPK4 (D) and AtSGT1b in *Nicotiana benthamiana*. Luciferase activity was determined with CCD camera. (E-F) Western blots shows protein accumulation in Fig 6G and 6H, S11C and S11D Fig. (G-H) RipAC does not associate with MPK3/4/6. The Split-LUC assay was performed as in Fig 2B and the RipAC-nLUC/cLUC-AtSGT1b pair was used as positive control. Protein accumulation was determined by western blot as shown in (H). These experiments were repeated at least 3 times with similar results. In western blot assays, protein marker sizes are provided for reference. (TIF)

**S12 Fig. MAPK-mediated phosphorylation of AtSGT1b contributes to *Ralstonia solanacearum* resistance in Arabidopsis.** (A, B) Soil-drenching inoculation assays in different Arabidopsis genotypes (*AtSGT1a-1* and its wild-type control Ws-0, *AtSGT1b-3* and its wild-type control La-*er*) were performed with GMI1000 WT strain. Composite data from 3 independent biological repeats (average values are shown in Fig 7A and 7B). n = 12 plants per genotype in each repeat in (A) and n = 15 plants per genotype in each repeat in (B). (C) Soil-drenching inoculation assays in Arabidopsis transgenic lines overexpressing *AtSGT1b* variants were performed with GMI1000 *ΔripAC* mutant. Composite data from 3 independent biological repeats. n = 15 plants per genotype in each repeat. (TIF)

**S1 Table. Resources used in this article.**
(DOCX)

**S2 Table. Primers used in this study.**
(DOCX)

## Acknowledgments

We thank Nemo Peeters and Anne-Claire Cazale for sharing unpublished biological materials, Yasuhiro Kadota and Ken Shirasu for helpful discussions and sharing biological materials, Suomeng Dong, Jian-Min Zhou, Gitta Coaker, Pengcheng Wang, and Laurent Deslandes for sharing biological materials, Rosa Lozano-Duran for critical reading of this manuscript, and Xinyu Jian for technical and administrative assistance during this work. We thank the PSC Cell Biology and Proteomics core facilities for assistance with confocal microscopy and LC-MS/MS analysis, respectively.

## Author Contributions

**Conceptualization:** Gang Yu, Alberto P. Macho.

**Data curation:** Gang Yu, Alberto P. Macho.

**Formal analysis:** Gang Yu, Alberto P. Macho.

**Funding acquisition:** Gang Yu, Alberto P. Macho.

**Methodology:** Gang Yu, Liu Xian, Hao Xue, Wenjia Yu, Jose S. Rufian, Yuying Sang, Rafael J. L. Morcillo, Yaru Wang.

**Project administration:** Gang Yu, Alberto P. Macho.

**Supervision:** Gang Yu, Alberto P. Macho.

**Visualization:** Gang Yu, Alberto P. Macho.

**Writing – original draft:** Gang Yu, Alberto P. Macho.

**Writing – review & editing:** Alberto P. Macho.

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
