## [Decision Letter · Decision Letter 0]

28 Apr 2020

Dear Dr. Macho,

Thank you very much for submitting your manuscript "A bacterial effector protein prevents MAPK-mediated phosphorylation of SGT1 to suppress plant immunity" for consideration at PLOS Pathogens. As with all papers reviewed by the journal, your manuscript was reviewed by members of the editorial board and by several independent reviewers. In light of the reviews (below this email), we would like to invite the resubmission of a significantly-revised version that takes into account the reviewers' comments.

We cannot make any decision about publication until we have seen the revised manuscript and your response to the reviewers' comments. Your revised manuscript is also likely to be sent to reviewers for further evaluation.

Sincerely,

Yuanchao Wang

Guest Editor

PLOS Pathogens

Wenbo Ma

Section Editor

PLOS Pathogens

Kasturi Haldar

Editor-in-Chief

PLOS Pathogens

orcid.org/0000-0001-5065-158X

Michael Malim

Editor-in-Chief

PLOS Pathogens

orcid.org/0000-0002-7699-2064

Reviewer's Responses to Questions

**Part I - Summary**

Reviewer #1: In this manuscript, the authors identified “A bacterial effector protein prevents MAPK-mediated phosphorylation of SGT1 to suppress plant immunity”. The story is of sufficient interest and novelty to warrant publication in PLoS Pathogens in my opinion. The authors showed a nice example that how plant pathogen Ralstonia solanacearum effector RipAC inhibits the interaction between SGT1 and MAP kinases, as well as the phosphorylation of a MAPK target motif in the C-terminal domain of SGT1. The presentation that RipAC suppresses ETI to manipulate plant immunity by targeting SGT1 in this manuscript is sorted in a clear logic. However, there are still some concerns need to be addressed before it can be accepted by the journal.

Reviewer #2: The NLR mediated immune response is important for recognizing pathogen effectors. The NLR resistance proteins, such as PRS2, Prf, Rx, RPS5 and RPM1…., can form a complex with other proteins (e.g. RAR1, require for Mla12 resistance; HSP90, heat shock protein 90; and SGT1, suppressor of the G2 allele of skp1) to regulate plant resistance by R protein accumulation and/or activation. Among these three proteins, several studies showed that bacterial effector (AvrBsT) and fungi effector (See1) can target SGT1 to regulate avrBsT-triggered cell death and DNA synthesis. In this study, the authors demonstrate that another effector from Ralstonia solanacearum, RipAC, also associates with SGT1 to regulate NLR (RPS2)-mediated immunity. The authors first showed that RipAC contributed to the virulence of R. solanacearum, then they further identified SGT1 as its target via MS analysis. In addition, SGT1 can associate with MPK4/6 and be phosphorylated by MAP kinases at two serine residues (NbSGT1 S282 and S358). RipAC reduced SGT1 phosphorylation via reducing the association of SGT1 and MPK6. Furthermore, SGT1 phosphorylation is required for RPS2-triggered cell death in N. benthamiana and RipAC virulence function in Arabidopsis. Overall, the work showed clear data that SGT1 is a target of RipAC and the virulence function of RipAC requires the phosphorylation of SGT1. Moreover, the manuscript is easy to follow and logic is clear. But I feel that some data do not directly support the conclusion, especially the part of SGT1 phosphorylation depends on MPK kinase (please see the comment 4). Also, the mechanism of SGT1 regulating NLR-mediated immunity is different, such as SGT1 stabilized NLR Rx and Prf (Azevedo et al, 2006, EMBO journal; Kud et al, 2013, Bio.&Bio Res. Commun.), but also destabilized RPS5 and RPM1 (Holt III et al, 2005, Science), so I think the authors should turn down some conclusions, like phosphorylation of SGT1 contributes to the activation of NLR-mediated responses in line 28-29. It might be better to use “RPS2-mediated response” instead of “NLR-mediated responses” when the authors make a conclusion. I think it will be worth considering to publish this study in Plos Pathogens if the authors can clarify the following questions.

1. The authors claimed that RipAC contributed to R. solanacearum infection in Arabidopsis expressing RipAC in Figure 1D, but not Pto in Figure 3G. Do the authors think that the plant defense suppressed by RipAC is general or specific?

2. In Figure 3H-J, expressing RipAC in Arabidopsis enhanced Pto AvrRpm1, Pto AvrRpt2, Pto AvrRps4 infection, how about the Arabidopsis HR phenotype in these leaves triggered by these effectors and R genes recognitioin? The authors should include these data in the figure to see whether RipAC affects HR.

3. co-IP, Split-LUC and split-YFP assays could not demonstrate direct interaction of RipAC with SGT1, just association. If the authors want to suggest directly interaction, in vitro pull-down assay is needed. Also, why the authors did not perform Split-LUC assay for SlSGT1 and RipAC? Considering SlSGT1 is also important for RipAC-mediated infection promotion in tomato, these data should be included.

4. The data supporting AtSGT1a/b phosphorylation by MPK3/4/6 are not sufficient and not direct in Figure 4D. The authors just used AvrRpt2 to activate MAPK kinase, then test whether AtSGT1a/b phosphorylation are affected. Several experiments could be used: 1) co-expression of SGT1a/b with MPK4, MPK6 (the authors used in Figure 6G) or MPK3 together with MKK5DD (constitutive activation of MPK3/6) or loss-of-function MKK5KR (constitutive inactivation of MPK3/6) (Guo et al, Nat. Comm., 2016) to see whether SGT1a/b phosphorylation was enhanced or reduced. 2) test SGT1a/b phosphorylation in mpk3, mpk4, mpk6 single mutants, or mpk3/mpk6 double mutant (silencing of mpk6 in mpk3 mutant, reported by Yu et al, Cell reports, 2019). 3) in vitro kinase assay for MPK3, MKK9DD and SGT1 as shown in Guo et al (Nat. Comm., 2016) to show that SGT1 is directly phosphorylated by MAP kinase. 4) use a AvrRpt2 mutants that can not induce MAPK activation to test SGT1a/b phosphorylation as a control in Figure 4D.

5. Did the authors test RPS2, Avr3a/R3 and RipE2-induced HR in SGT1-silenced plants in N. benthamiana? Also, how about RipAC suppressed RPS2, Avr3a/R3 and RipE2-induced HR in SGT1-silenced plants in N. benthamiana?

6. RipAC reduced AtSGT1a/b phosphorylation in Arabidopsis expressing RipAC in Figure 6A. Was AtSGT1a/b phosphorylation affected in Arabidopsis and tomato after infected by R. solanacearum?

7. The author claimed that RipAC inhibited the SGT1-MPK3/4/6 interaction, resulting in reduced SGT1 phosphorylation in Figure 6G, 6E, 6F. However, in Figures 3C, 3D, 3E, 6C, RipAC also reduced the MPK3/6 phosphorylation. So is it possible that RipAC reduces the phosphorylation of MPK3/4/6 resulting in a decrease in SGT1 phosphorylation? Did the authors test whether RipAC can interact with MPK3/4/6?

8. SGT1 2D phosphorylation enhanced RPS2-triggered cell death in Figure 5A-5C in N. benthamiana. How about the cell death after co-expression SGT1 2D or SGT1 DD with RipAC in N. benthamiana?

9. The association of SGT1 2D mutant with RPS2 is reduced in Figure 5D, but the RPS2-mediated cell death is enhanced by SGT1 2D in Figure 5A. As the authors mentioned, the enhanced SGT1 phosphorylation might lead to RPS2 to be released from the complex. Just curious, is there any literature reporting that SGT1 phosphorylation is directly involved in the accumulation, recognition and activation of NLR protein?

10. SGT1 phosphorylation is very important for RPS2-mediated cell death and RipAC virulence; and the two phosphorylation sites (S271, T346) located in the SGS region, which is required for interaction with RipAC. So how about the interaction of these two mutants with RipAC compared to the interaction of SGT1 WT with RipAC?

11. Figure 5B, 5C should include the results of SGT1 AA and DD induced cell death.

12. It seems that knockout of SGT1a or SGT1b in Arabidopsis did not affect Pst DC3000 infection (Azevedo et al, 2006, EMBO journal). How about R. solanacearum?

**Part II – Major Issues: Key Experiments Required for Acceptance**

Reviewer #1: Major concerns:

1) The authors present that an interaction of AtSGT1s with AtMAPK3 and AtMAPK6，and MAPK-mediated phosphorylation is important for SGT1 function. Do AtMAPK3 and AtMAPK6 directly phosphorylate AtSGT1?

2) If AtSGT1 is phosphorylated by AtMAPK3 and AtMAPK6，you could knockout or silence AtMAPK3/AtMAPK6 and NbMAPKs, and test SGT1-mediated immune responses and phosphorylation level in SGT1 in Arabidopsis and N. benthamiana, respectively.

3) The authors show that independent transgenic lines expressing RipAC exhibit enhanced disease symptoms upon soil-drenching inoculation with R. solanacearum. All the data only use WT Col-0 as control, it’s better to find a RipAC mutant or EV as control for the experiments.

4) The authors present that Co-localization of plasma membrane-associated CBL-GFP protein and SGT1-RFP in N. benthamiana (Fig S5C). But it appears that it is difficult to distinguish the precision localization of plasma membrane and the cytoplasm. You could determine their precision localization through wall separation experiments.

Reviewer #2: (No Response)

**Part III – Minor Issues: Editorial and Data Presentation Modifications**

Reviewer #1: Minor concerns

1) Line 230, change Agrobacterium-mediated transformation to Agrobacterium-mediated transient expression.

2) Fig S5F, change cLUC+SGT1b to cLUC-SGT1b.

3) It’s better to use Empty vector as control in Fig 5B.

4) Line 154, from a 35S constitutive promoter to driven by single or double 35S constitutive promoter.

5) Fig 3G-J, I don’t know if SGT1-independent NLR protein exists. If it exists, It’s better to use SGT1-independent NLR as negative control.

6) Fig1,in infection experiments, no difference analysis of the data, so the virulence function of RipAC is not accurately proved.

7) Figure 2A,the Western results of Flag IP is not full picture.

8) Figure 2B, Split-LUC assay does not have quantification of the luciferase signal.

9) In the process of verifying the interaction between SGT1 and MAPKs, the authors have different results from different methods.

Reviewer #2: (No Response)

PLOS authors have the option to publish the peer review history of their article (what does this mean?). If published, this will include your full peer review and any attached files.

Reviewer #1: No

Reviewer #2: No
---

## [Decision Letter · Decision Letter 1]

27 Aug 2020

Dear Alberto,

We are pleased to inform you that your manuscript 'A bacterial effector protein prevents MAPK-mediated phosphorylation of SGT1 to suppress plant immunity' has been provisionally accepted for publication in PLOS Pathogens. Congratulations!

Please note that the reviewers suggested some minor editorial changes. Please consider and revise accordingly during the publication stage.

Best regards,

Wenbo Ma

Section Editor

PLOS Pathogens

Wenbo Ma

Section Editor

PLOS Pathogens

Kasturi Haldar

Editor-in-Chief

PLOS Pathogens

orcid.org/0000-0001-5065-158X

Michael Malim

Editor-in-Chief

PLOS Pathogens

orcid.org/0000-0002-7699-2064

Reviewer Comments (if any, and for reference):

Reviewer's Responses to Questions

**Part I - Summary**

Reviewer #1: The authors have fully addressed all of my concerns.

Reviewer #2: In this revised manuscript, the authors have addressed most of my questions. This work clear show that RipAC targets SGT1 to suppress its phosphorylation, thereby attenuating plant immunity. The study further highlights the importance of effectors on pathogen virulence, and also expand the profiles of SGT1 in plant immunity. I think this work is suitable for for publication in “Plos Pathogens” after revision.

**Part II – Major Issues: Key Experiments Required for Acceptance**

Reviewer #1: (No Response)

Reviewer #2: no

**Part III – Minor Issues: Editorial and Data Presentation Modifications**

Reviewer #1: Figure 2A, should add FLAG before all SGT1s and should provide a better image of AtSGT1a IP results.

Figure 2C，why there is a weak band when RipAC was incubated with immobilized GST-GUS.

Figure 4，change SGTs to SGT1s.

Figure 5B, change camera to visible light.

Reviewer #2: For Figure 3 results, the authors need to correct the conclusion that RipAC inhibits SGT1-dependent immune response and the description in the results section, because there is no direct data to support this conclusion. The data in Figure 3 only suggest that RipAC inhibits effector triggered HR and immunity.

PLOS authors have the option to publish the peer review history of their article (what does this mean?). If published, this will include your full peer review and any attached files.

Reviewer #1: No

Reviewer #2: No

---

## [Editor Report · Acceptance letter]

21 Sep 2020

Dear Dr. Macho,

We are delighted to inform you that your manuscript, "A bacterial effector protein prevents MAPK-mediated phosphorylation of SGT1 to suppress plant immunity," has been formally accepted for publication in PLOS Pathogens.

Best regards,

Kasturi Haldar

Editor-in-Chief

PLOS Pathogens

orcid.org/0000-0001-5065-158X

Michael Malim

Editor-in-Chief

PLOS Pathogens

orcid.org/0000-0002-7699-2064